# SimiC enables the inference of complex gene regulatory dynamics across cell phenotypes

Jianhao Peng [1,12], Guillermo Serrano [2,12], Ian M. Traniello[3,4], Maria E. Calleja-Cervantes [2,5], Ullas V. Chembazhi [6], Sushant Bangru [6], Teresa Ezponda [5,7], Juan Roberto Rodriguez-Madoz [5,7], Auinash Kalsotra [3,6,8], Felipe Prosper [5,7,9], Idoia Ochoa [1,10,11✉] & Mikel Hernaez [2,3,7,11✉]

Single-cell RNA-Sequencing has the potential to provide deep biological insights by revealing complex regulatory interactions across diverse cell phenotypes at single-cell resolution. However, current single-cell gene regulatory network inference methods produce a single regulatory network per input dataset, limiting their capability to uncover complex regulatory relationships across related cell phenotypes. We present SimiC, a single-cell gene regulatory inference framework that overcomes this limitation by jointly inferring distinct, but related, gene regulatory dynamics per phenotype. We show that SimiC uncovers key regulatory dynamics missed by previously proposed methods across a range of systems, both model and non-model alike. In particular, SimiC was able to uncover CAR T cell dynamics after tumor recognition and key regulatory patterns on a regenerating liver, and was able to implicate glial cells in the generation of distinct behavioral states in honeybees. SimiC hence establishes a new approach to quantitating regulatory architectures between distinct cellular phenotypes, with far-reaching implications for systems biology.

[1] Department of Electrical and Computer Engineering, University of Illinois at Urbana-Champaign, Urbana, IL 61801, USA. [2] Computational Biology Program, CIMA University of Navarra, IdiSNA, Pamplona, Spain. [3] Carl R. Woese Institute for Genomic Biology, University of Illinois at Urbana-Champaign, Urbana, IL 61801, USA. [4] Neuroscience Program, University of Illinois at Urbana-Champaign, Urbana, IL 61801, USA. [5] Hemato-Oncology Program, CIMA University of Navarra, IdiSNA, Pamplona, Spain. [6] Department of Biochemistry, University of Illinois at Urbana, Urbana, IL 61801, USA. [7] Centro de Investigación Biomédica en Red de Cáncer, CIBERONC, Madrid, Spain. [8] Cancer Center@Illinois, Urbana, IL 61801, USA. [9] Hematology and Cell Therapy, Clinica Universidad de Navarra, Pamplona, Spain. [10] Department of Electrical Engineering (TECNUN), University of Navarra, San Sebastian, Spain. [11] Data Science and Artificial Intelligence Institute (DATAI), Universidad de Navarra, Pamplona, Spain. [12] These authors contributed equally: Jianhao Peng, Guillermo Serrano.
✉email: idoia@illinois.edu; mhernaez@unav.es

Gene regulatory networks (GRNs) infer regulatory interactions between genes, including relationships between transcription factors and their targets, and have become one of the most important steps in determining cellular functions[1,2] and modeling different systemic behaviors[3,4]. In addition, GRNs have been found to be reliable surrogates/predictors of behavioral state[5,6]. GRNs are usually represented as graphs, where the nodes are genes and the edges represent a regulatory (or co-expression) relationship between the genes that they connect. These graphs can be broadly classified as: directed, if the regulatory direction is known; weighted, where the weight of each edge represents the regulatory strength of the connection; or bipartite, where genes are split into disjoint sets and edges only connect genes of distinct sets. In addition, some GRN inference methods follow a module-based approach, where genes are first clustered in modules and then a GRN is inferred per module, in contrast to other methods that build a unique single GRN for the data[7,8].

Until recently, most available gene expression data were derived from "bulk" RNA-Sequencing (RNA-Seq), which does not differentiate among the cellular composition of a tissue sample, and therefore only gives an average measure of the gene expressions across all cells. More recently, single-cell RNA-Sequencing (scRNA-Seq) has made it possible to acquire gene expression data for individual cells in samples containing up to millions of cells[9]. scRNA-Seq data are generally summarized as a matrix containing the expression values of genes for each sequenced cell. Available bioinformatic pipelines then group these cells by similarities in gene expression patterns, thus generating cell clusters that are predicted to represent distinct cell types that share anatomical or functional characterization[10,11]. However, the nature of scRNA-Seq data limits the applicability of traditional GRN inference methods to single-cell expression data. In particular, GRN inference methods for bulk RNA-Seq data assume that gene expressions across samples are independent and identically distributed (i.i.d.). However, this does not apply to single-cell data, as cells in the same cluster exhibit similar expression patterns.

The newly developed computational methods for scRNA-Seq data provide a new landscape for single-cell GRN inference, as the data can be imputed to reduce its intrinsic high sparsity caused by the dropout effect[12–17], and the information on cell types, such as their pseudo-temporal ordering, can be used as side information for GRN inference[18,19]. For example, the ordering of cells is used to infer the regulatory relationships among genes via ordinary differential equations[20], correlation methods[21], information theoretic measures[22], or boolean functions[23]. To control the dropout effect, Papili et al.[24] used time-stamped scRNA-Seq data and transformed the expression data into (ordered) distances between expression profiles before inferring the relationships among genes using ridge regression, whereas Yuan et al.[25] converted the expression profiles into images and used additional side information to train a deep convolutional neural network for inference of gene relationships. Although not its primary goal, the single-cell GRN inference method SCENIC[26,27] generates scores indicative of the activity of a transcription factor (TF) and its associated target genes (denoted as a *regulon*) in a particular cell, thus providing in principle, the capability to study the regulatory dynamics of regulons across cells. Similarly, ICAnet[28] provides activity scores for each of the modules (i.e., set of associated genes) it infers. However, ICAnet module score may refer to several TFs. Furthermore, a given TF may be encountered in several modules simultaneously, hindering the inference of regulon-level regulatory dynamics. For a comprehensive survey on GRN inference methods for scRNA-Seq data, see[29].

All the GRN inference methods discussed above produce a unique GRN per input dataset. However, related cells, such as those arising on a cell differentiation path, will be potentially characterized by similar, but distinct enough, GRNs, as it is natural to assume that there should be a smooth transition between the GRNs associated with each phenotype[30]. Similarly, the GRNs of identical cell types under different conditions are expected to vary, and even slight changes can be informative regarding the effects of specific cellular conditions. In this context, directly inferring GRNs independently for each cell phenotype might result in a group of divergent networks that share little in common. It would therefore be useful to add constraints on the inferred GRNs to allow for comparisons of GRN architectures across related cell types in different conditions, as it would allow capturing the underlying complex gene regulatory dynamics driving the transitions between cell phenotypes.

We present SimiC, a GRN inference algorithm for scRNA-Seq data that takes as input single-cell imputed expression data, a list of driver genes, the cell labels (cell phenotypes), and the ordering information, and produces a GRN for each of the different phenotypes. Given the provided ordering between the cell phenotypes, SimiC adds a similarity constraint when jointly inferring the GRNs for each phenotype, ensuring a smooth transition between the corresponding GRNs. Evaluating the quality and accuracy of GRNs is not a straightforward task, as there are no clear metrics for evaluation[13,19]. Besides, most existing GRN inference algorithms only give a static picture of the GRN and provide hypotheses based on such network. SimiC, on the other hand, generates multiple networks at the same time, providing a framework to further evaluate the network dynamics. Hence, drawing from Aibar et al.[26], we propose a *regulon activity score* that captures, per driver gene, the activity of their associated gene set in each individual cell. We further evaluate changes in distribution of the activity scores across cell phenotypes to describe the *regulatory dissimilarity* of each driver gene across cell phenotypes.

All these analyses have been incorporated into the SimiC framework, and we show, on both simulated and real datasets, that SimiC enables the inference of complex gene regulatory dynamics across cell phenotypes. In particular, we show on simulated data that SimiC's inferred GRNs obtain higher micro-F1 and Cohen's kappa coefficients, as well as better goodness of fit as measured by the adjusted $R^2$. We also test SimiC on a variety of real datasets, including model and non-model organisms, and show that SimiC, contrary to SCENIC[26,27], ICAnet[28], and SINCERITIES[24], was able to recapitulate complex gene regulatory dynamics and accurately characterize them via the proposed *regulon activity* and *regulatory dissimilarity* scores. These methods are considered the state-of-the-art. Note also from the review in[29] that most available methods cannot handle more than 2000 genes. From those that do, SCENIC is the best performing one, and the only one to generate a set of regulons that allows the user to assess dynamics at the single-cell level. In addition, SCIMITAR[31] is no longer maintained and we were not able to run it; SCODE[20] is designed to work with TF-TF interactions and is not recommended for full GRN inference; LEAP[21] generates a gene-gene correlation matrix at specific "lags", thus making it unfit to generate regulon activity scores across cell-types.

Taken together, we show the importance of being able to jointly infer several GRNs from related cell phenotypes (such as different timepoints). The joint GRN inference performed by SimiC enables capturing regulatory dynamics across cell phenotypes that are missed by previously proposed GRN inference methods.

## Results

**Overview of the method**. We consider imputed scRNA-Seq expression data with the corresponding cell labels that indicate

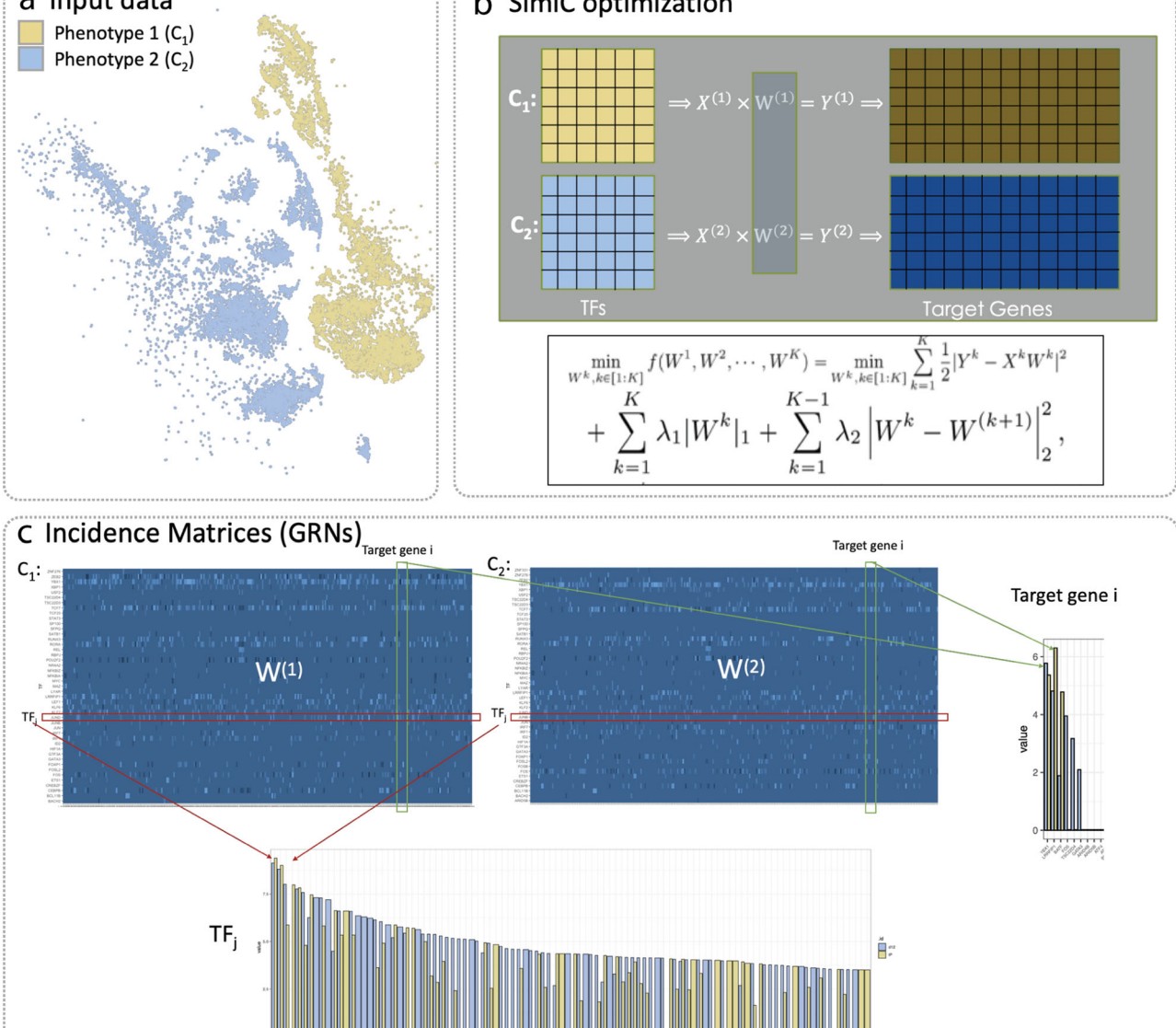

**Fig. 1 Workflow of SimiC. a** Example of input data to SimiC consisting of the imputed cell expression data of cells with phenotype 1 ($C_1$) and phenotype 2 ($C_2$). The data is visualized in the tSNE-reduced space, where the color indicates the cell phenotype. **b** SimiC objective function based on LASSO[32] with an added similarity constraint. The $W^{(i)}$s represent the inferred incident matrices, and the $X^{(i)}$s and $Y^{(i)}$s the expression matrices of the driver and target genes, respectively. The superscripts correspond to the cell phenotypes ($K$ in total). $\lambda_1$ and $\lambda_2$ are hyperparameters to control the weights of the regularization terms. **c** Incidence matrices inferred by SimiC, one per phenotype. The dimension is the number of driver genes, such as transcription factors (TFs), by number of target genes. Weight $W_{i,j}$ represents the influence of the $i$th TF in the $j$th target gene. The bar plots show the differences in the inferred weights for a given TF and target gene for the two phenotypes (indicated by color).

the phenotype of each cell (and possibly additional information such as cell type), as well as a linear ordering between cell phenotypes (Fig. 1a). In cases with only two cell phenotypes (e.g., case vs control), the ordering is implicit. Note that when phenotype is not available, the ordering of cells can be inferred, for example, from the pseudotime of cells using pseudotime inference methods.

SimiC expects the scRNA-Seq data to be represented as a matrix of dimension number of genes by number of cells, in which the set of driver genes are specified. We note that in our analyses we use transcription factors (TFs) as the driver genes. SimiC draws from the fused LASSO regression technique[32] to infer a GRN for each cell phenotype, while imposing a (learned) level of similarity across GRNs of contiguous cell phenotypes (see

"Methods"). Thus, the output of the optimization problem is a set of incidence matrices representing the GRNs associated to each cell phenotype (Fig. 1b). These inferred matrices are of dimension number of TFs by the number of target genes, and each entry is a weight representing the influence of the TF in the corresponding target gene (Figs. 1c and 2a). In what follows, and following the nomenclature of[26,27], we denote *the regulon of a TF* as its connected target genes. However, unlike Aibar et al.[26], we take into account the weights of the edges between the target genes and the TF. Factoring the weights allows us to account for both the strength and the direction (promotion or repression) of the regulation between the TFs and their associated target genes.

In order to evaluate the regulatory dissimilarity of the different TFs across the different cell phenotypes, we perform the following

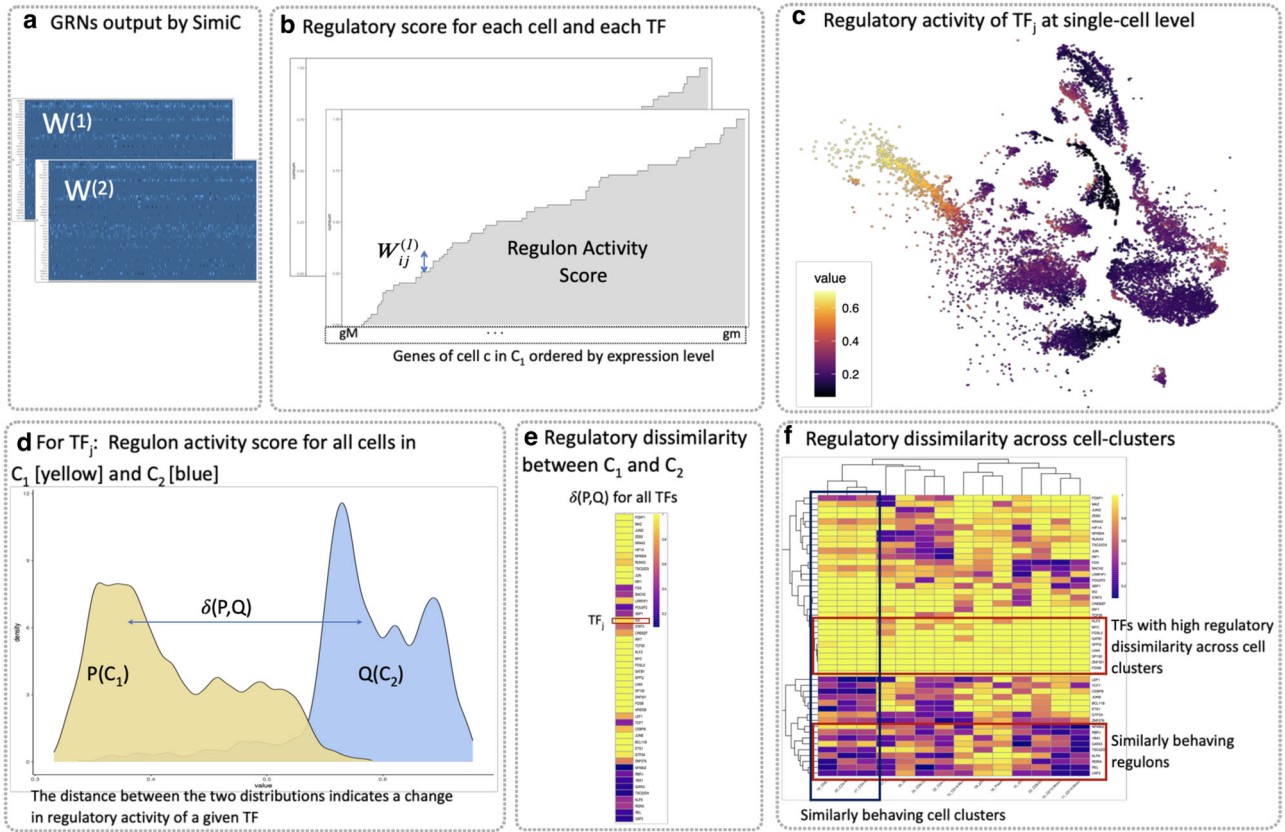

**Fig. 2 Proposed metrics to unravel complex gene regulatory dynamics. a** SimiC's inferred incidence matrices $W^{(1)}$ and $W^{(2)}$ for phenotype 1 and 2, respectively, for the input data of Fig. 1a. The incidence matrices correspond to the inferred gene regulatory networks (GRNs). **b** For a given regulon, and hence transcription factor (TF), we compute the regulon activity score per cell as follows. First, for a given cell $c$, we sort the target genes by their expression values in an increasing order. Then, for each sorted target gene we show the contribution of its corresponding weight in the weighted regulon by increasing the y-axis by that amount. The regulon activity score corresponds to the area under the generated curve. **c** tSNE plot visualizing the input scRNA-Seq data, where each cell is colored by its regulon activity score for $TF_j$. **d** For a given TF, using the cells' individual regulon activity score, we can compute the empirical regulon activity score distribution for each cell phenotype. An example of the obtained distributions for the considered two phenotypes and a given TF are depicted, $P(C_1)$ for phenotype 1 (yellow) and $Q(C_2)$ for phenotype 2 (light blue). **e** Computing the distance between the obtained distributions for each TF provides a metric for the regulatory dissimilarity of each TF between the cells belonging to phenotype 1 ($C_1$) and those belonging to phenotype 2 ($C_2$). The heatmap shows an example of the obtained regulatory dissimilarity for several TFs when the total variation distance $\delta(P,Q)$ is used. **f** The dissimilarity score can be further computed for different cell-clusters across the phenotypes, as shown in the heatmap, which reveals TFs with high dissimilarity scores across cell clusters as well as TFs with similar ones.

analysis on the inferred GRNs. First, we measure how the activity of the different regulons change across cell clusters. We propose to use a new metric to compute it: the *regulon activity score* (Fig. 2b, "Methods"), inspired by the AUC score from[26,27]. For each cell, the regulon activity score serves to quantify the relative activity of a given regulon with respect to the expression of all target genes. Intuitively, for a given TF, if the most expressed target genes in a cell correspond to those with the highest weights in its weighted regulon, the TF in question may have a large influence on the expression profile of the cell. This is represented by a large regulon activity score. The regulatory activity of a regulon can also be visualized at single-cell resolution in the tSNE-reduced space, giving more insight into how a specific TF changes its activity at single-cell level (Fig. 2c).

For each TF, we also compute the empirical distribution of the regulon activity score for each cell phenotype (Fig. 2d). We then compute the distance (via the total variation metric) between empirical distributions to measure how the activity of the regulons change across different cell phenotypes (Fig. 2e, "Methods"). We refer to this distance as the *regulatory dissimilarity*. Further, subdividing the overall cell population into smaller clusters allows us to deepen into the regulatory

dissimilarity across cell phenotypes at cell-cluster level. Thus, the SimiC workflow can generate a heatmap that shows the regulatory dissimilarities for all regulons and all cell clusters. This heatmap allows us to uncover shifts in regulatory activity that are associated with different conditions, environments, or developmental states (Fig. 2f).

Finally, when assessing the accuracy of the inferred GRNs, we measure the goodness of fit via the adjusted $R^2$ coefficient between the true target genes' expression and the one computed as the linear combination of the TFs' expressions inferred by SimiC (via the GRNs). Specifically, we split the data into training and testing sets with a 80/20 ratio (note that this split is only done for assessing the accuracy, the remaining results are obtained from applying SimiC to the whole data). We then infer the GRNs on the training data, and use the inferred networks, together with the TFs' expression on the test data, to predict the target genes' expression. Finally, we compute the adjusted $R^2$ on the test data using the true target genes' expression and the predicted one. In addition, the sparsity of the regulatory program of each target gene (that is, the set of TFs regulating a given target gene) must be also controlled, as otherwise, the LASSO optimization may generate highly dense GRNs, which is undesirable[7,33]. Thus, on

real datasets, we control the sparsity via hyperparameter tuning to achieve desirable levels of sparsity while maintaining satisfactory goodness of fit (see "Methods").

SimiC is implemented in Python but we also provide a script to call the program from R. The code and scripts are available at https://github.com/jianhao2016/SimiC.

**SimiC correctly captures network dynamics on synthetic data.** We first evaluated the performance of SimiC on synthetic data satisfying the assumption that GRNs change smoothly between states, and showed that it captured the network dynamics more accurately than the previously proposed inference method SINCERITIES[24]. To the best of our knowledge, SINCERITIES is the only existing method that also takes as input scRNA data sequenced at different timepoints (or that belong to different phenotypes) and connects TFs and target genes with weighted edges, which allows for a fairer comparison. Note, however, that SINCERITIES generates a unique GRN as output, contrary to SimiC, which generates a GRN per phenotype. Specifically, we did not consider SCENIC[26,27] for this comparison as it can not be run without transcriptome annotations (such as TF-binding sites), which do not apply in this case. In addition, the GRN inference methods GENIE3[34] or GRNBOOST2[35], which are used internally by SCENIC, do not produce weighted GRNs, which impairs their analysis based on model fitting. These same reasons prevented us from using ICAnet[28] in this analysis.

We simulated scRNA-Seq data for five different cell states (or phenotypes) with a linear order between them, where the underlying GRNs of consecutive states did not change abruptly (Fig. 3a, see Methods). We simulated scRNA-Seq data for 1000 cells per state, for a total of 5000 cells. We considered 50 TFs and 20 target genes, and GRNs containing 3 types of edges: no regulation (edge weight = 0), activation (edge weight = +1) and depression (edge weight = −1). Finally, we simulated the TF's expression by sampling from a negative binomial distribution[36,37], and computed the targets' expression using the constructed GRNs and the TF's expressions, alongside an added Gaussian noise. Specifically, for a given cell and state, the expression of the $j$th target gene $y_j$ is generated as $\sum_i W_{i,j} x_i + n_i$, where $W_{i,j}$ represents the GRN of the considered state, $x_i$ represents the expression of the $i$th TF (for the same cell), and $n_i$ is modeled as Gaussian random noise (Fig. 3a, see "Methods").

We ran SimiC and SINCERITIES on the generated synthetic data and, to showcase the importance of the state separation in fitting the data, we also ran LASSO on all states combined. Note that in the latter case only one GRN was generated. For each method, we computed the micro-F1 score and the Cohen's kappa coefficient of the inferred networks. The micro-F1 score assesses the quality of multi-label binary problems and measures the F1 score of the aggregated contributions of all classes. The Cohen's kappa coefficient is more robust in multi-class prediction and provides an insight into how well the model performs when compared to a random guess. Since the constructed GRNs only contain edges equal to 0, +1, or −1, we set a threshold (0.5 by default) such that all weights below −0.5 are converted −1, all weights above 0.5 to +1, and the rest to 0.

SimiC obtained the highest micro-F1 score for all states as compared to SINCERITIES and LASSO (Fig. 3b). The difference in performance is more pronounced with SINCERITIES. Whereas SimiC obtains micro-F1 scores close to 1, SINCERITIES obtains micro-F1 scores around 0.8. Due to the sparsity of the GRNs, the micro-F1 score is highly biased towards the F1 score of predicting 'no edges' (0 weight) and does not accurately reflect the capability of the method to predict the other two types of edges (+1 or −1). The Cohen's kappa coefficient does not suffer

from this, as it measures the gain of the model when compared to a random guess based on the support of each type of edges. Whereas SimiC achieved Cohen's kappa coefficients around 0.8, SINCERITIES's scores oscillate around 0 and did not surpass 0.075 (Fig. 3c). With respect to LASSO, we observed that it produced Cohen's kappa coefficients below 0.8 in all states. We also observed that the Cohen's kappa coefficient for SimiC was stable with the threshold used to convert the weights into +1, −1, and 0, similar to what is observed with LASSO (Supplementary Fig. 1a, b). However, this was not the case for SINCERITIES (Supplementary Fig. 1c). These results demonstrate that when SimiC's assumptions are met, SimiC better captures the states' network dynamics, as it infers more accurate GRNs than SINCERITIES and LASSO. We also observed that, as expected, the GRNs inferred by SimiC do not change abruptly across consecutive states, reproducing the simulated GRNs (Fig. 3d, e).

To assess the goodness of fit of SimiC and evaluate the effect of $\lambda_2$ ($\lambda_2$ represents the weight assigned to the similarity constraint in SimiC's optimization problem, see Methods), we computed the adjusted $R^2$ coefficient for values of $\lambda_2$ ranging from $10^{-5}$ to 1, while the rest of the hyperparameters are kept constant. We observed that the average adjusted $R^2$ for SimiC ranged from about 0.8, achieved with the smallest value of $\lambda_2$, to about 0.85, achieved with the largest value of $\lambda_2$ (Fig. 3f). Hence, in this case, putting more weight in the similarity constraint produced more accurate GRNs, which led to more accurate expression for the target genes. Note that this was expected, as the synthetic dataset satisfies SimiC's assumptions in regards to similarity across GRNs of consecutive states. As a comparison, removing the similarity constraint (i.e., setting $\lambda_2 = 0$, which corresponds to LASSO run independently in each state) yielded an adjusted $R^2$ of about 0.835, which corresponds to a reduction of about 2%. On the other hand, running LASSO on the combined data yielded an adjusted $R^2$ of approximately 0.49. Overall, these results show that jointly inferring a GRN per state while introducing a similarity constraint better captures the underlying network dynamics when a smooth transition between states exist.

In terms of running time, SimiC employed 4 min to infer the GRNs and 3 min to compute the regulon activity scores, while SINCERITIES employed 2 min to infer the GRNs (see "Methods").

**ChIP-Seq data validate a high proportion of the TF-target connections inferred by SimiC.** To assess the performance of SimiC on real datasets, we first evaluated the correctness of the association between the TFs and their corresponding target genes. Two scRNA-Seq datasets (GEO accession ID GSE139369) were considered for the analysis: (i) a scRNA-Seq dataset composed of monocyte cells coming from either bone marrow (BM, 2681 cells) or peripheral blood (PB, 3610 cells)[38]; and (ii) a scRNA-Seq dataset composed of CD4+ T-lymphocytes cells also coming from either BM (5297 cells) or PB (7563 cells)[38]. For both datasets, after pre-processing the data (see "Methods"), we selected the 100 most variant TFs and the 1000 most variant target genes, where the variation was measured as the mean absolute deviation (MAD).

On the dataset composed of monocyte cells, running SimiC resulted in two GRNs, each representing the regulatory landscape of monocytes in PB or BM. SimiC obtained a median adjusted $R^2$ on test data of 0.861 (Supplementary Fig. 2a), while yielding a median of 7 TFs regulating each target gene (Supplementary Fig. 2f). Next, we analyzed in more detail the regulatory relationships inferred by SimiC between TFs and target genes using ChIP-Seq data. For each TF, we sought empirical evidence of TF binding sites up to 5 Kbp upstream of its target genes. We

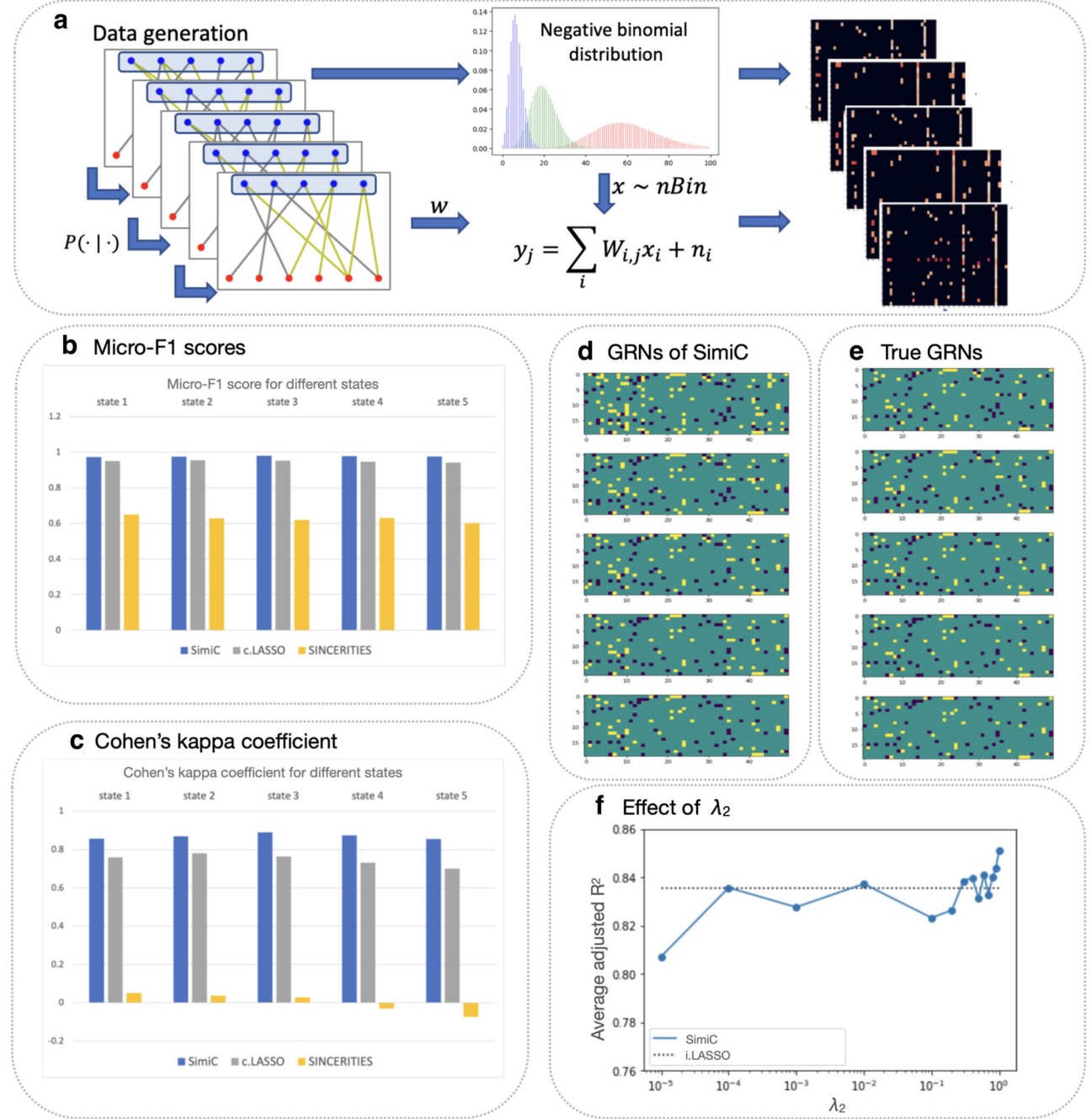

**Fig. 3 SimiC accurately infers network dynamics on synthetic data. a** Workflow for generating the synthetic data. First, we generate the regulatory network of each state such that no abrupt changes occur between the networks of two consecutive states. We then generate the TF's expression data by sampling from a negative binomial (*nBin*) distribution[36, 37]. Finally, we generate the target genes' expression data using the generated networks and the expression of the TFs. In particular, the expression of the $j$th target gene $y_j$ is generated as $\sum_i W_{i,j} x_i + n_i$, where $W_{i,j}$ is extracted from the generated incidence matrix, $x_i$ represents the expression of the $i$th TF, and $n_i$ is modeled as Gaussian random noise (independent and identically distributed across samples). **b** Micro-F1 score and **c** Cohen's kappa coefficient on edge prediction obtained by SimiC, c.LASSO (LASSO run in all states combined), and SINCERITIES. State-of-the-art GRN inference methods SCENIC and ICAnet were left out of the analysis as they need additional information not available for synthetic data. **d** Obtained incidence matrices (GRNs) for SimiC and **e** simulated (true) incidence matrices. **f** Effect of $\lambda_2$ on the adjusted $R^2$ obtained with SimiC. The adjusted $R^2$ obtained with independent LASSO (i.LASSO) is included for comparison (note that $\lambda_2 = 0$ in this case).

extracted the TF binding sites information from the ReMap2020 database[39]. Out of the 100 considered TFs, 7 were dropped as no target genes were assigned to them (due to potential colinearities with other TFs) and 28 lacked ChIP-Seq experiment data, and hence they were also excluded from the analysis. Out of the remaining 65 TFs, 47 had their set of target genes heavily enriched in the corresponding set of TF binding sites (odds ratio

[OR] > 2, *p*-value < 0.05, see "Methods"), while the remaining TFs showed partial ChIP-Seq evidence (Supplementary Table 1 and Supplementary Data 1). Note that SimiC infers regulons based solely on the gene expression profiles from the input scRNA-Seq data. Therefore, some edges inferred by SimiC may be due to an indirect regulation between the TF and the target gene. In such cases, as the TF is likely not binding in the proximity of

the target gene, it may not be possible to validate the regulatory interactions via the performed ChIP-Seq analysis.

When focusing on the differences in the composition of the weighted regulons across tissues (i.e., BM vs PB), we observed, for example, a change in the composition of the regulon of KLF4, a well-known transcription factor involved in essential monocyte development from the monocyte-dendritic cell progenitors[40]. Thus, as expected by its importance in monocyte development, the regulon of KLF4 had larger weights on the BM than on PB (Supplementary Fig. 3a). In addition, KLF4 had, exclusively on BM, connections to target genes *MGST1*, *ICAM3*, and *S100A12*, all playing a crucial role during monocyte differentiation and maturation[41–43].

We performed a similar analysis on the CD4+ T-lymphocytes cells coming from either BM or PB[38]. SimiC also generated two GRNs in this case, which exhibited a median adjusted $R^2$ of 0.840 on test data (Supplementary Fig. 2b) while yielding a median of 6 TFs regulating each target gene (Supplementary Fig. 2g). The ChIP-Seq analysis revealed that out of the 63 TFs with ChIP-Seq evidence, 34 had their associated target genes enriched (OR > 2, *p*-value < 0.05, Supplementary Table 2, Supplementary Data 1), while the other 29 presented partial ChIP-Seq evidence (Supplementary Table 2, Supplementary Data 1). When delving into specific regulons, we found that *RUNX3*, a TF involved in the activation of naive CD4+ T cells during immune response[44], was positively associated with target genes *HLA-F*, *PTPN6*, *GADD45B*, *SFRS9,* and *SEMA4D*, all key genes on the stimulation and activation of CD4+ T cells[45–49] (Supplementary Fig. 3b).

**SimiC unravels complex regulatory dynamics of CD8+ CAR T cells during immunotherapy of non-Hodgkin lymphoma patients**. We assessed the capabilities of SimiC to uncover complex gene regulatory dynamics of engineered CD8+ T lymphocytes during immunotherapy of non-Hodgkin lymphoma patients. The development of adoptive cell therapy approaches based on T lymphocytes engineered with chimeric antigen receptors (CAR T cells) has been a breakthrough for cancer immunotherapy strategies[50,51]. CAR T cells generated by current technologies are enriched in different types of long-lived memory cells that can quickly expand to large numbers of terminally differentiated effector T cells upon exposure to their antigen. However, after long-term antigen stimulation CAR T cells can become exhausted, leading to a defective or insufficient antitumoral function. As functionality of CAR T cells depends on an orchestrated activation of specific signaling pathways, understanding the changes in gene regulatory dynamics suffered by the infused CAR T cells after exposure to their cognate antigen is of utmost importance for the efficient design and application of this immunotherapy strategy.

In this context, Sheih et al.[52] generated a single-cell transcriptome dataset of CD8+ CAR T cells isolated from the infusion product (IP), as well as from peripheral blood at the expansion peak after treatment (days 7–14; termed D12) of patients with relapsed and refractory non-Hodgkin lymphoma (GEO accession ID GSE125881). This dataset contains 7616 cells for the IP phenotype and 8576 cells for the D12 phenotype, and we selected the most variant 100 TFs and 1000 target genes for the analysis. We applied SimiC on the scRNA-Seq data to infer two akin GRNs governing the CAR T cell function at IP and D12 timepoints (median adjusted $R^2$ of 0.834 on test data, Supplementary Fig. 2c; median of 5 TFs regulating each target gene, Supplementary Fig. 2h).

To understand the changes in the overall regulatory changes between IP and D12, we computed the *Kleinberg authority score*[53] for each target gene of each GRN (see "Methods"). In brief, an authority represents a target gene that is regulated by many different TFs, which, at the same time regulate many target genes. SimiC was able to capture a shift in the overall authority scores after tumor recognition by CAR T cells, indicating an increment in the regulatory activity (Fig. 4a, Wilcoxon rank test, FDR < 0.05). Specifically, we observed that the authority score of key genes in tumor recognition, such as *GZMB*, *NOSIP* and multiple *HLA* family genes, were significantly incremented in D12. These genes have been widely studied as effectors of the acute response of the immune system[54,55].

A specific example of such regulatory shifts between IP and D12 arose on the network spanned from antagonist regulons MYC[56] and RUNX3[57]. We observed that their shared targets had opposite weights, such as *GZMB*, which at IP was regulated positively by RUNX3 and negatively by MYC (Fig. 4b). Further, at the expansion peak (D12), the negative regulation from MYC was lost, while the positive one from RUNX3 was maintained. Another example of such reversed regulation between MYC and RUNX3 is the target gene *SELL*, which controls the proliferation of memory cells and was positively regulated by MYC and negatively by RUNX3[58]. Finally, we note that these shifts in connectivity between the different timepoints were shared across most regulons as the authority scores of target genes were generally increased on D12 (Fig. 4a).

In order to delve into the regulatory dynamics of individual TFs across cell phenotypes, we computed the distribution, across all cells, of the *regulon activity scores* (see Methods). Recall that, for a given TF, this score measures the relative activity of the set of target genes connected to the TF on a given cell. Due to their special relevance at different stages of the immune response, we first focused on the TFs *MYC*, *RUNX3,* and *EOMES*, genes with an antagonist behavior[56,57,59]. The regulon of the proto-oncogene *MYC* showed its regulatory peak activity at the IP, where different peaks could be observed yielding a multimodal distribution, while it showed an overall lack of activity at the expansion peak (D12) (Fig. 4c). This was expected due to the *MYC* role in cellular proliferation, a process that is partially halted on cells carrying out an increased effector role in the immune response[60]. The regulon of the *RUNX3* gene, which is required to maintain a lytic activity once the cells have met the antigens[44], showed a multimodal distribution with a heavy tail at D12, as majority of cells showed little activity of the RUNX3 regulon while other cells showed high activity (Fig. 4d). As we will show in the next section, this multimodality and heavy tail arose primarily due to the different levels of exhaustion reached by the CAR T cells at D12. The EOMES regulon showed, at D12, high activity on majority of cells, while, similar to RUNX3, EOMES is innactive at IP (Fig. 4e). Note that a heavy tail on the distribution at D12 is also observed for EOMES, where some cells at D12 showed little activity. This behavior is explained as *EOMES* progressively promotes exhaustion on anti-tumor CD8+ cells[61].

To examine in more detail the activity of these regulons at single-cell level, we colored the input single-cell data by their respective regulon activity scores and visualized it in a tSNE plot (Fig. 4f–h). For the MYC regulon, we observed a cluster of MYC-active cells at IP at the bottom right side, and a bottom-up gradient along which the regulon activity decreased progressively (Fig. 4f, top). On the other hand, almost all cells at D12 were MYC-innactive, with the exception of a small cluster of cells at the top-left-most edge (Fig. 4f, bottom). Interestingly, these cells happened to be a specific subtype of CAR T cells which were transitioning from memory CAR T cells to exhausted CAR T cells, as further described in the following section. When focusing on the RUNX3 regulon, cells were mostly RUNX3-innactive at IP (Fig. 4g, top), while we observed a clear left-to-right gradient of RUNX3 activity on D12 (Fig. 4g, bottom). For

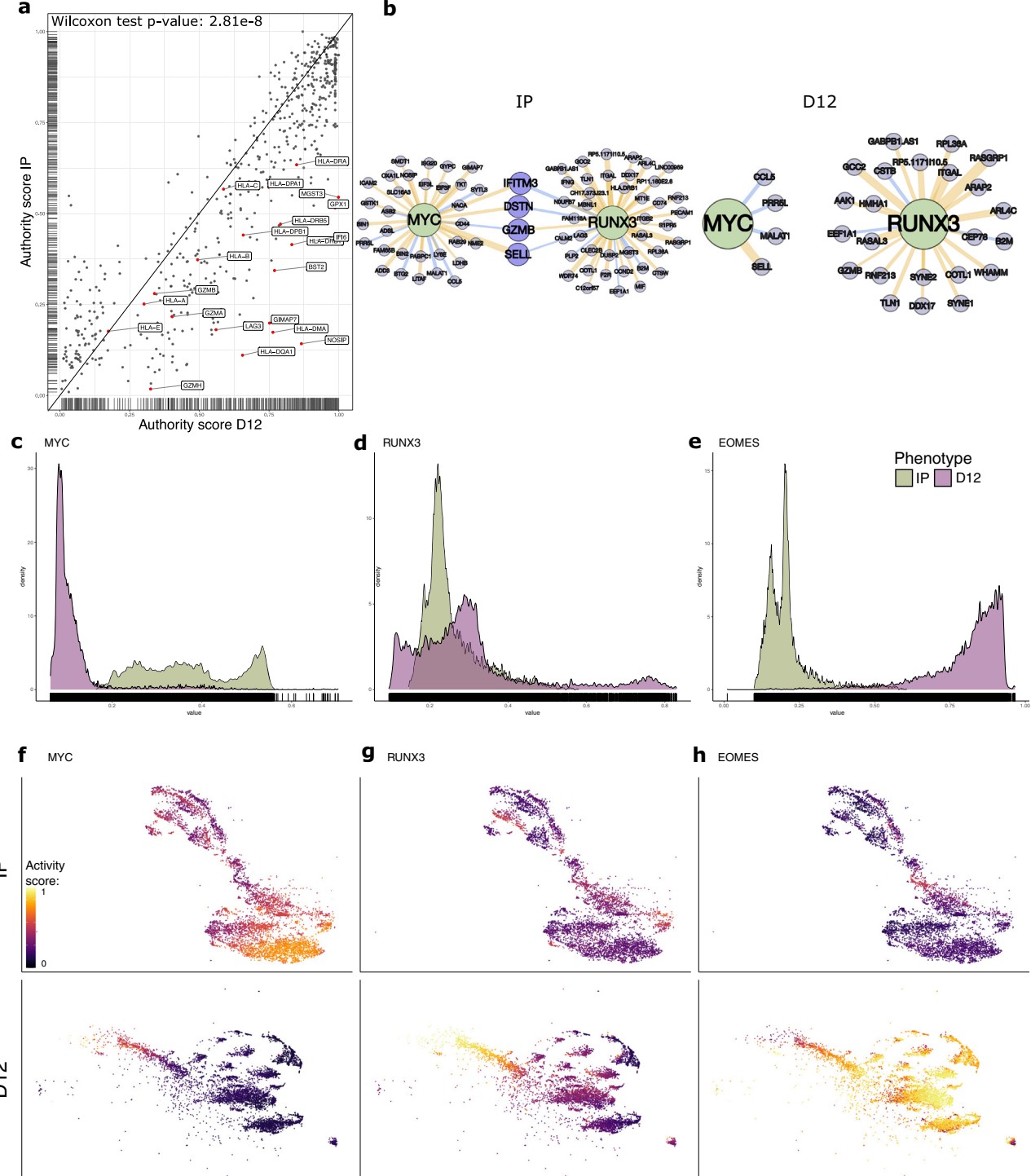

**Fig. 4 SimiC captures complex regulatory dynamics within CAR T cells (CD8+ CAR T cell dataset[52]). a** Comparison of the authority scores for each target gene in the two phenotypes IP and D12. We observe an increment of the authority score in D12 as compared to IP (Wilcoxon rank test, FDR < 0.05). **b** Network representing the weighted regulons of the transcription factors MYC and RUNX3 for the considered phenotypes IP and D12. The color of the edges linking the TFs with their corresponding target genes indicates whether the regulation is positive (yellow) or negative (blue), and the width of the edge corresponds to its strength (i.e., the inferred weight). Regulon activity score distribution for the two different phenotypes IP and D12 for MYC (**c**), RUNX3 (**d**), and EOMES (**e**) regulons. tSNE plots showing the CAR T cells at IP (top) and D12 (bottom) colored by their MYC (**f**), RUNX3 (**g**), and EOMES (**h**) regulon activity score. The peak of activity of the MYC regulon is observed at IP, while at D12 it gets progressively turned off. The RUNX3 regulon has its peak of activity after tumor recognition at D12, while its activity gets turned off once cells get exhausted at D12. Finally, the EOMES regulon gets more active once CAR T cells get exhausted after tumor recognition on D12.

the EOMES regulon we observed, at IP, a behavior similar to that of the RUNX3 (Fig. 4h, top). However, at D12, we observed a nearly-opposite behavior to RUNX3, where cells got progressively activated following the same left-to-right gradient (Fig. 4h, bottom). These gradients were explained by the progressive exhaustion suffered by the CAR T cells, as T cell exhaustion is promoted by EOMES while inactivates RUNX3 (see next section).

Next, we compared the capacity of SimiC to uncover phenotype-specific regulatory activity to that of SCENIC and ICAnet. SINCERITIES was left out of the analysis as it needs at least four states (phenotypes) to run. Since both methods (SCENIC and ICAnet) infer a unique GRN for a given input, we ran them on the whole dataset, as well as on the IP and D12 cells independently. Note that in the first case only one GRN was inferred for all cells, while in the latter case two GRNs were inferred, each capturing the regulatory dynamics of the IP or D12 cells. One of the difficulties we encountered is that some of the analyzed TFs (MYC, EOMES, and RUNX3) were not retained by these methods. Furthermore, when run on each phenotype independently, there is no guarantee that the same set of TFs will be retained in both cases. For example, when run on the whole data, SCENIC retained MYC and EOMES, but not RUNX3. The same occurred when run independently on each phenotype, except that EOMES was not retained in D12. With ICAnet, the comparison becomes even more complicated, as it generates several modules of related genes, and a given TF can be contained in several modules simultaneously. With ICAnet, only RUNX3 was retained, and when run independently on each phenotype, RUNX3 was contained in 39 modules for IP and in none for D12.

The scores computed by SCENIC and ICAnet did not show phenotype-specific regulatory dynamics when run jointly on IP and D12 or independently (Supplementary Fig. 4). In particular, contrary to the regulon activity score distributions obtained by SimiC for the MYC regulon (Fig. 4c), the AUC distributions obtained by SCENIC showed a large overlap, resulting in no clear distinction between the cells on each phenotype (Supplementary Fig. 4a, b). Note also that, contrary to SimiC, no gradient was observed across the cells of neither of the phenotypes. For the EOMES regulon, when run jointly, almost no differences were observed in the AUC scores between the cells of the two phenotypes (Supplementary Fig. 4c), contrary to SimiC, which captured the expected dynamics (Fig. 4h). Running SCENIC on IP independently produced similar AUC scores as when run jointly (Supplementary Fig. 4d).

When running ICAnet jointly on the data, we found one module containing TF RUNX3. However, the produced scores did not show any notable differences across the cells belonging to the IP and D12 phenotypes (Supplementary Fig. 4e). When run independently, there were 39 modules containing TF RUNX3 on IP, and none on D12 (Supplementary Fig. 4f). Most of the 39 modules produced scores showing a gradient from top to bottom on the cells at IP, with the bottom cells being more active. This gradient was more pronounced on some modules (Supplementary Fig. 5), and only few modules showed all cells being mainly inactive, similarly to the scores produced by SimiC (Fig. 4g, top).

In terms of running time, SimiC employed 32 min to infer the GRNs, SCENIC 9 min, and ICAnet 132 min (see "Methods"). The reported time for SCENIC and ICAnet is when run on the whole data.

**SimiC captures differences in regulatory activity within the different CD8+ CAR T cell sub-populations through tumor recognition.** While GRNs are generally computed on cells coming from similar lineages or phenotypes[24,26], specific regulons of these networks may be differently activated depending on the current state of the cell. For example, differential activation of regulons has been shown to play a key role on differentiating cells[27,38].

In what follows, we show, using the dataset from the previous section, that SimiC is able to capture the underlying regulatory activity within the different CAR T cell subpopulations and unravel the differences in their activity after tumor recognition. Further, we show that SCENIC and ICAnet were not able to properly capture these dynamics. In addition, SimiC yielded improved clustering performance of cell states, as measured by the adjusted rand index (ARI) scores[62].

Once the GRNs for phenotypes IP and D12 using the whole CAR T cell population were inferred by SimiC (see previous section), and the regulon activity score computed for each cell, we grouped cells by their state; namely, CD8+ memory, CD8+ effector and CD8+ exhausted cells. To annotate each cell, cells were first clustered using Seurat[10], yielding 8 cell clusters that were then merged into three clusters using a curated set of marker genes associated to each of the above-mentioned cell states (Supplementary Data 2). We then computed for each cell state and each TF the distribution of regulon activity score, which yielded two empirical distributions, one for each phenotype (IP and D12). Finally, we computed the distance between these distributions and denoted it as the *regulatory dissimilarity score* (see "Methods"), which measures, for each TF and cell state, the differences in overall regulon activity between phenotypes IP and D12 (Fig. 5a).

The multimodal distributions observed in the regulon activity of RUNX3 and MYC (Fig. 4c, d) are also reflected in the heatmap of Fig. 5a, where RUNX3 and MYC (among others) show different levels of regulatory dissimilarity scores for different cell populations. Specifically, a low regulatory dissimilarity score was observed on exhausted CAR T cells for RUNX3 while high dissimilarity can be observed in the same population for MYC. We were also able to uncover a set of TFs showing high regulatory dissimilarity between IP and D12 across all CAR T cell subpopulations (Fig. 5a, yellow cluster). TFs yielding such high regulatory dissimilarity scores regardless of the cell subtype are expected to modulate the general activity of the immune response after tumor recognition. Example of such TFs are ID2 and BATF genes, key transcriptional regulators essential in the immune response of the CD8+ T cells[63,64]. These TFs presented lower values of regulon activity on the IP cells, and a clear increment on their regulon activity after tumor recognition (D12) (Supplementary Fig. 3c, d).

We next show that SimiC, contrary to SCENIC and ICAnet, was able to recapitulate key CAR T cell regulatory dynamics at cell-state resolution. Using SimiC, we observed that MYC, a proliferation marker, had its highest regulon activity on the memory cells, due to the proliferation potential of these cells, while EOMES, an exhaustion promoter, was highly inactive (Fig. 5b, f). As the immune activity progressed, the cells gradually became effector-like carrying the immune response (CD8+ effector cells). Hence, the proliferation is gradually halted diminishing the activity of the MYC regulon and progressively increasing the activity of RUNX3, which is required to maintain lytic activity once cells have met the antigens (Fig. 5b, d). Once the effector cells transitioned towards the exhaustion state, the activity of the MYC and RUNX3 regulons got halted, as the proliferative and effector capabilities of cells were being lost, reaching their lowest values on the CD8+ exhausted cells (Fig. 5b, d). On the other hand, the activity of EOMES got progressively increased, reaching its peak at fully exhausted cells (Fig. 5f). These subtle, albeit important, dynamics of regulon activities were recapitulated by neither the SCENIC AUC scores nor ICAnet module scores (Fig. 5c, e, g).

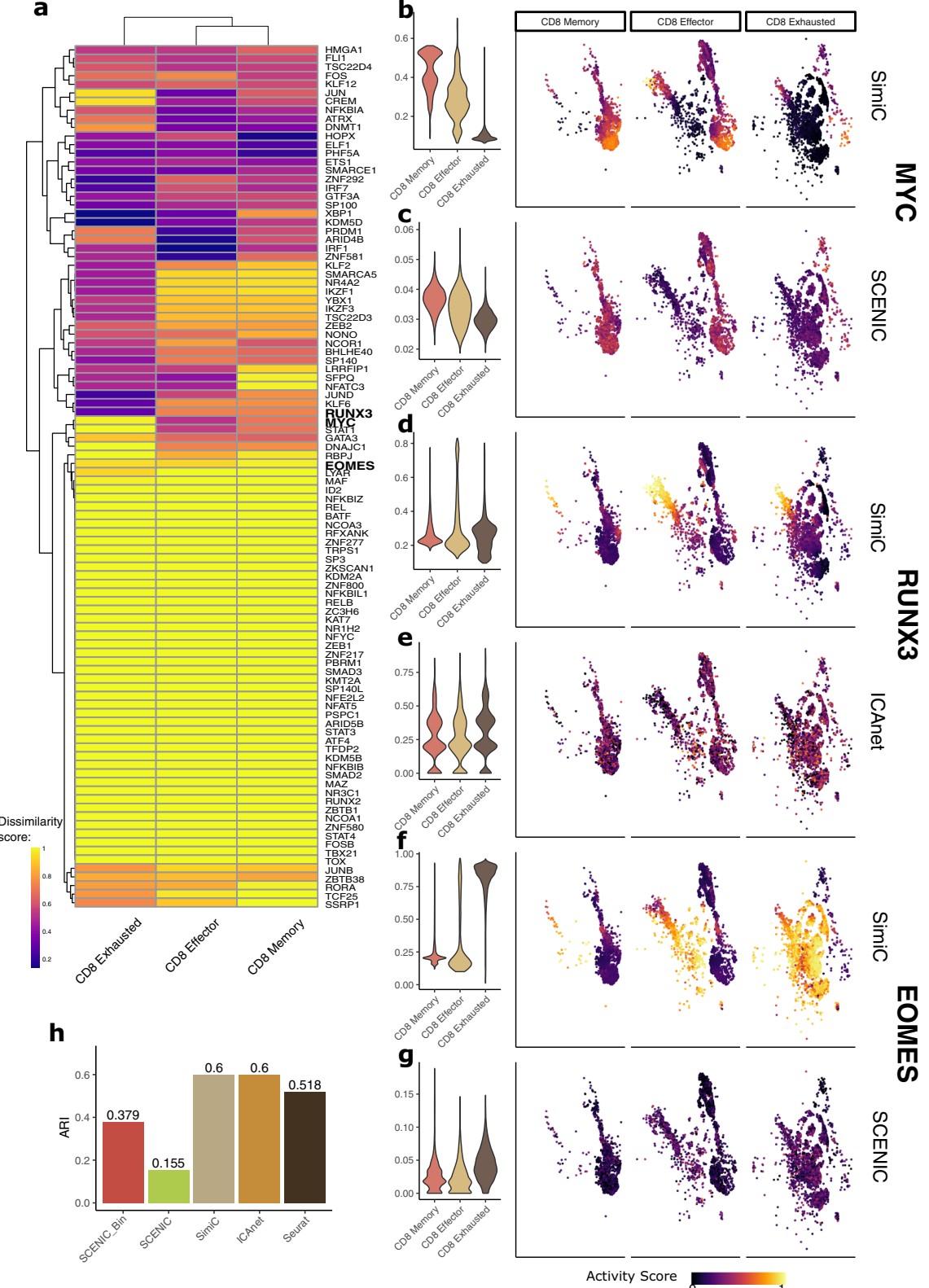

**Fig. 5 SimiC's proposed scores capture cell-state specific regulatory dynamics on the CD8+ CAR T cell dataset[52]. a** Heatmap depicting the regulatory dissimilarity between the two cell phenotypes (IP and D12) for different regulons for CD8+ exhausted, CD8+ effector, and CD8+ memory cells. Violin plots of the activity scores for the different cell states, as well as the tSNE plots showing the CAR T cells of different cell states colored by their activity score for the MYC regulon as calculated by SimiC (**b**) and SCENIC (**c**); for the RUNX3 regulon as calculated by SimiC (**d**) and ICANet (**e**); and for the EOMES regulon as calculated by SimiC (**f**) and SCENIC (**g**). **h** Cell state clustering performance as measured by the ARI score. For SCENIC we consider the binarized and non-binarized scores. A larger ARI indicates a better clustering performance. SINCERITIES was left out of the analysis as it needs at least four phenotypes to run.

To quantify the state-specific clustering capabilities of the three considered methods, we compared the clustering capabilities of SCENIC and ICAnet (when run on the whole data) when using their dedicated clustering methods, to that of SimiC when its activity scores are clustered using k-means. For SCENIC, we considered both the binarized and non-binarized AUC scores. For reference, we also considered directly clustering the expression data using Seurat. In all cases, three clusters were generated. SimiC yielded the highest ARI score among all considered scenarios (Fig. 5h), meaning that SimiC's activity scores captured better the differences between the cell states CD8+ memory, CD8+ effector, and CD8+ exhausted. Finally, we analyzed the capacity of the non-binarized AUC and module scores from SCENIC and ICANet, respectively, to uncover phenotype- or state-specific regulatory activity (via hierarchical clustering). Neither of them (SCENIC and ICAnet) showed clear phenotype-specific clusters when run jointly on IP and D12, nor state-specific clusters when run jointly or on each phenotype independently (Supplementary Fig. 6a–d). This is in contrast to SimiC, whose regulon activity scores clustered cells by phenotype, and showed partial clustering by cell-states (Supplementary Fig. 6e), manifesting the capacity of SimiC to uncover activity scores related to the phenotype at hand.

To further validate the ability of SimiC to cluster cells based on the generated activity scores, and compare it to that of SCENIC and ICAnet, we performed an additional experiment with the CD4+ T-lymphocytes dataset[38]. Specifically, we selected five known trajectories related to cell-types, namely: (i) HSC -> Early Erythrocyte -> Late Erythrocyte, (ii) HSC -> CLP -> NK, (iii) HSC -> CLP -> B, (iv) HSC -> CLP -> CD8+ CD4+, and (v) HSC -> GMP -> Monocytes CD14+. We applied SimiC to the cells of each trajectory (using the corresponding ordering), and then clustered the cells with k-means ($k = 3$) using the generated regulon activity scores. Given the resulting clustering (rand idx (RI) scores close to 1 in all cases), we conclude that SimiC's inferred GRNs and their corresponding regulon activity scores can reliably differentiate cells (Supplementary Fig. 7). We performed a similar analysis using SCENIC and ICAnet, when run independently on each cell type and when run on the whole dataset. The results show that SCENIC's computed regulons and corresponding scores are able to capture the cell-states (in four out of the five trajectories) when run independently on each state, but not when run jointly (Supplementary Fig. 7). The metrics generated by ICAnet, in both considered cases, are not able to differentiate cells by their state (Supplementary Fig. 7).

**SimiC uncovers key regulatory mechanisms of a regenerating liver across several time-points at cell-state resolution.** We next show how SimiC was able to model complex regulatory dynamics on datasets with multiple sequential cell phenotypes (i.e., timepoints). To this end, we used an scRNA-Seq dataset (GEO accession ID GSE151309) generated from regenerating mouse livers, which contains timed gene expression profiles of around 12K hepatocytes[65]. Specifically, 2/3rd partial hepatectomy was performed and cells were sequenced at different timepoints (24h, 48h, and 96h) post-surgery. Healthy adult mouse liver cells were also sequenced, yielding a total of four timepoints, termed adult, PHx24, PHx48, and PHx96. Non-hepatocyte cells were identified and removed from the dataset. The cells were annotated by their functional state as quiescent, proliferative, metabolically hyperactive, or transitioning[65].

Running SimiC on this dataset generated four akin GRNs, each representing the regulatory landscape at a given timepoint (median adjusted $R^2$ of 0.885 on test data, Supplementary Fig. 2d; median of 7 TFs regulating each target gene, Supplementary

Fig. 2i). When the number of cell phenotypes is larger than two, we define the regulatory dissimilarity score to be proportional to the area between the largest and the smallest densities at each regulon activity value (see "Methods"). This score therefore yields a dissimilarity score of 1 for non-overlapping distributions and a score of 0 for completely overlapping distributions. For example, TF YBX1 presented highly overlapping regulon activity densities, yielding a dissimilarity score of 0.41 on proliferating cells; while the densities of the TF CEBPB were mostly non-overlapping across all cell types, notably yielding a dissimilarity score of 1 for proliferating cells (Fig. 6a, b).

Two important regulatory patterns arise in the modeling, via GRNs, of the regenerating liver[65]. On the one hand, regulons following an initiation-progression pattern typically show low activity values in adult hepatocytes, while after hepatectomy there is a peak of activity (PHx24 and PHx48); then, the original value is regained again at PHx96. On the other hand, regulons following a termination-rematuration pattern show high activity in adult hepatocytes, while post-hepatectomy there is a consistently less activity (PHx24 and PHx48); and the activity rises again at PHx96. Two well known representatives of these patterns are the key hepatocyte TFs CEBPB (initiation-progression pattern) and CEBPA (termination-rematuration pattern), transcription factors that bind to similar DNA sequences[66] and play divergent but key roles in liver regeneration[65,67]. These activity patterns were clearly represented by the distribution of the CEBPA and CEBPB regulon activity scores across the different timepoints (Fig. 6c). Specifically, CEBPB had its activity peak at the first stages of liver regeneration after the hepatectomy (PHx24 and PHx48), and was then progressively being reduced towards the values of the adult cells. Opposite to this drift, the maximum activity of CEBPA was on the adult cells, and dropped to its minimum just when the regeneration of the tissue starts (PHx24), and later recovered its activity when the tissue converges to a mature state.

Next, we split the activity scores at each timepoint by the different cell states (Fig. 6d). We observed that almost no transitioning or proliferating cells were present in the adult liver, likely due to its mature state. Interestingly, we observed that on the course of liver regeneration, the activity of regulon CEBPA was similarly distributed on both transitioning and proliferating cells while its activity was regained faster on quiescent cells (Fig. 6d, top). On metabolically hyperactive cells, the activity density of CEBPA was bimodal, with some cells regaining CDBPA activity at faster pace than others (Fig. 6d, top). In the process of recovering the intermediate metabolic functions, metabolically hyperactive cells activated CEBPA at different paces, as activation of metabolic routes on hepatocytes depend, among other things, in the complex crosstalk between *CEBPA* and other important TFs such as *FOXA1* and *HNF4A*[68,69], which led to bimodalities in the CEBPA regulon activity density. Similarly, other regulons following a termination-rematuration pattern such as HNF4A and ONECUT2, which have more mature functionalities[68], were more active during the quiescent and metabolically hyperactive states[65,68] (Supplementary Fig. 8b).

An analogous behavior was observed on the regulon activity distribution of CEBPB, as well as on other regulons with an initiation-progression pattern like YBX1 and FOXA3[65,69], although following the opposite pattern (Fig. 6d, bottom, and Supplementary Fig. 8a), as they showed higher activity during proliferation compared to other states. Importantly, the regulatory dissimilarity score correctly captured the observed differences in activity across timepoints in a single score, allowing the clustering of TFs that have similar regulatory dynamics (Fig. 6a).

Further, we identified a correlation between the regulon activity and the pseudo-temporal transition of the hepatic cells

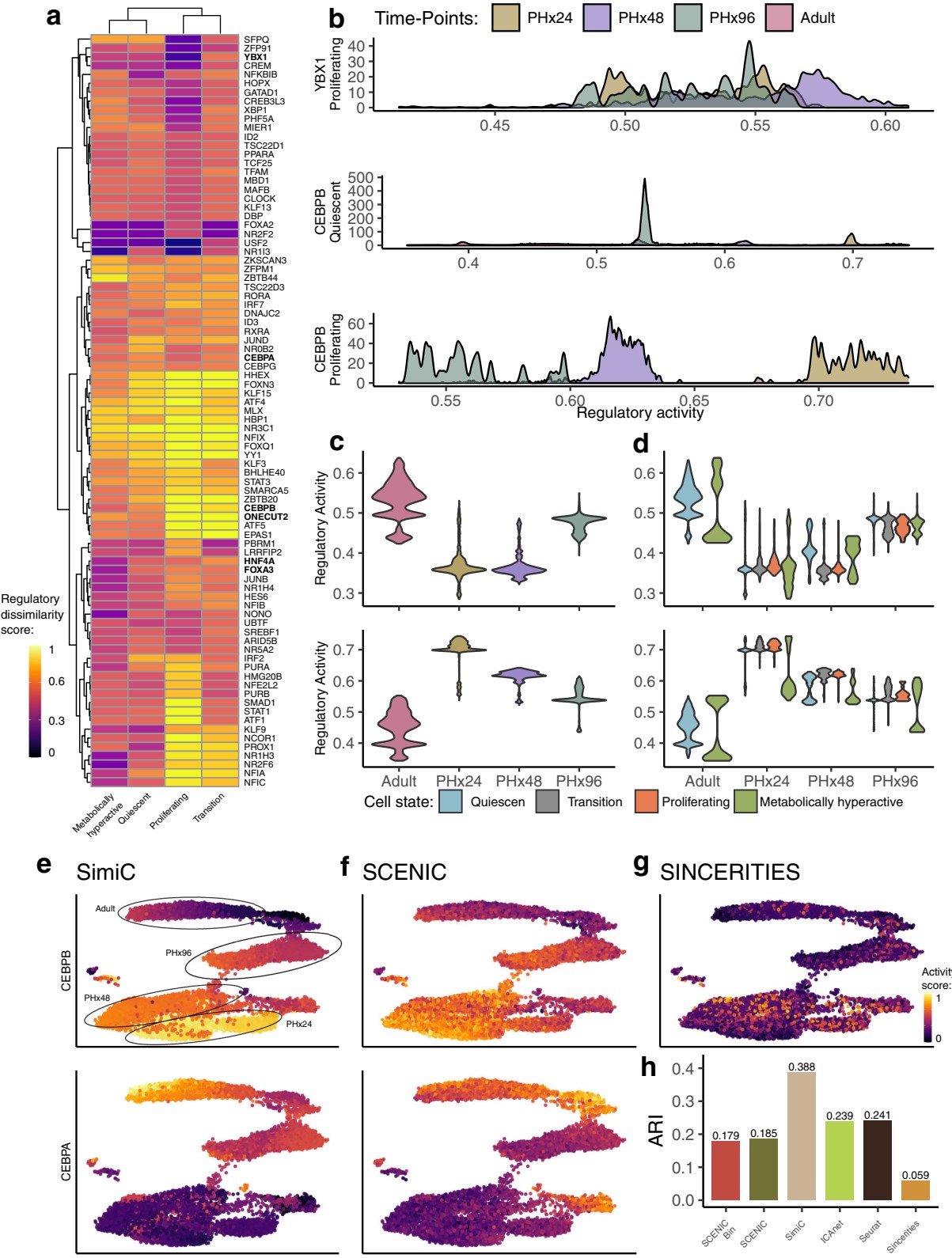

at single-cell resolution (Fig. 6e). Specifically, we observed a clear gradient of regulon activity for CEBPA, which had very low activity at early stages of regeneration (PHx24 and PHx48) and its activity progressively increased while cells progressed towards adult liver cells (Fig. 6e, top). The opposite behavior could be observed for the activity of regulon CEBPB, whose activity progressively decreased as the liver regenerated (Fig. 6e, bottom).

These results show that SimiC is able to capture the regulatory dynamics of a regenerating liver at single-cell resolution.

Note that a similar analysis was performed by Chembazhi et al.[65] using SCENIC. However, SCENIC cannot jointly compute different GRNs for all timepoints, and therefore, a unique GRN was computed using all data points. In the original analysis, while SCENIC was able to capture similar patterns, the activity

**Fig. 6 SimiC's inferred GRNs across timepoints on a regenerating liver (hepatocyte dataset[65]) capture regulatory dynamics of hepatocytes in different functional states. a** Heatmap of the regulatory dissimilarity score of quiescent, proliferating, metabolically hyperactive, and transitioning cell types across the four sequential timepoints for the different regulons, showing cell and time dependent variation. **b** Density plots showing the regulatory activity for the regulon YBX1 on the proliferating cell population and for the regulon CEBPB on the quiescent and proliferating cell populations, for the four considered timepoints. Violin plots showing the distribution of the regulatory dissimilarity scores for regulons CEBPA (top) and CEBPB (bottom) across timepoints (**c**) and across cellular states of the liver regeneration, for the four considered timepoints (**d**). tSNE plot showing the hepatocyte cells, colored by their regulatory activity score as computed by SimiC (**e**), SCENIC (**f**), and SINCERITIES (**g**) for the regulons CEBPB (top) and CEBPA (bottom). No results for CEBPA with SINCERITIES are shown as this TF did not appear in the inferred GRN. Similarly, ICAnet did not generate any modules with the CEBPA or CEBPB TFs. Timepoint to which each cell belongs is also specified. **h** Cell state clustering performance as measured by the ARI score. For SCENIC we consider the binarized and non-binarized scores. A larger ARI indicates a better clustering performance.

differences captured by SCENIC across the different timepoints yielded smooth and unimodal distributions (Fig. 4 of Chembazhi et al.[65]). The smooth distributions yielded by SCENIC are a consequence of computing a unique GRN for all cells and timepoints, which intrinsically averages out the different regulatory patterns across timepoints. On the other hand, the distributions yielded by SimiC across all cells were generally multimodal, which showcases the capability of SimiC to capture the differences in regulatory behavior across cell types. In addition, the AUC scores from SCENIC or SINCERITIES were not able to recapitulate the mentioned patterns at single-cell resolution as accurately as SimiC (Fig. 6f, g). We also run ICAnet on this dataset, but none of the generated modules contained any of the analyzed TFs (YBX1, CEBPA, or CEBPB).

Finally, when clustering the activity scores across all cells, we showed that SimiC captured the cell states better than SINCERITIES, SCENIC, and ICAnet (the latter two when run on the whole data rather than on each phenotype separately), as measured by the ARI score (Fig. 6h). As in the analysis of the CAR T cells, we used their dedicated clustering methods for SCENIC and ICAnet, and k-means for SimiC and SINCERITIES. Four clusters were generated in all cases, since there are four cell states (quiescent, transitioning, proliferating, and metabolically-hyperactive). In addition, when applying hierarchical clustering to SimiC's activity scores of all cells within a given timepoint, the cells belonging to the same state generally clustered together (Supplementary Fig. 8c). Thus, SimiC was able to uncover the main differences in regulatory dynamics across timepoints for different cell states, offering deeper biological insights than previously available.

**SimiC relates a key honeybee transcription factor to aggressiveness and associates it to Glia cells.** We examined the performance of SimiC on non-model organism lacking highly curated annotations. Some of the previously proposed methods for scRNA-Seq GRN inference, such as SCENIC[26] or ICANet[28], rely on additional annotations of the transcriptome (such as motifs or TF-binding site coordinates in SCENIC or Protein-Protein interaction networks in ICANet) to perform the GRN inference, and hence they are not suitable for non-model organisms.

We applied SimiC to an existing dataset that provides single-cell brain transcriptomic data generated in western honey bees (*Apis mellifera*, GEO accession ID GSE130785). In the accompanying study, Traniello et al.[70] performed scRNA-Seq on a whole-brain (WB) and mushroom body (MB) sample collected from two female honey bees. The whole-brain sample contained all brain structures, including sensory neuropils like the antennal lobe (AL), optic lobe (OL), and mushroom bodies (MB), a higher-order sensory processing region composing roughly 1/3rd of the bee brain[71] (Fig. 7a). In contrast, for the MB sample, all brain

regions except the MB were removed prior to sequencing. Further, cells were clustered using Seurat[10], yielding 11 cell clusters[70] (Fig. 7b). Transcription factors were obtained from[5,72]. We utilized these data to see if SimiC could identify regulatory activity that is unique to cell types captured in the whole-brain but not in the mushroom body sample. After generating the two GRNs with SimiC (one for MB and one for WB, median adjusted $R^2$ on test data of 0.834, Supplementary Fig. 2e; median of 3 TFs regulating each target gene, Supplementary Fig. 2j), we computed the regulatory dissimilarity score between MB and WB cells, for the different cell clusters.

The cell clusters coming from the OL, the primary visual neuropil of the honey bee brain and also one of the largest and most molecularly diverse regions[73], clustered together by the dissimilarity score across TFs (Fig. 7c, right-hand side). These cell clusters were identified by the expression of OL-specific markers like *scr*, *gad1*, *drgx*, and *SoxN*[74–76], and others (Supplementary Data 3), and represent neuronal subtypes exclusive to the insect visual system. Furthermore, OL neurons 3, the OL cell cluster specifically annotated as lobular T4/T5 neurons based on the expression of *drgx*, *SoxN*, and *slo*[75–78], yielded one of the highest overall dissimilarity scores. Besides, MB-specific neuronal subtypes, identified by expression of *Mblk-1*[79,80] and *CaMKii*[81], also clustered together (Fig. 7c, left-hand side).

Further, we found that the Drosophila melanogaster orthologue of the immune signaling gene *Deaf1* (LOC412296) had more regulatory activity in the MB compared to WB (Fig. 7d). In support of our findings, Deaf1 has been specifically identified in the honey bee MB as a regulator of aggression[72]. Furthermore, we found higher *Deaf1* activity in glia compared to the rest of the clusters (Fig. 7e). Glia makes up only a small fraction of the insect brain but have been hypothesized to play a role in regulating the neurometabolic state underlying aggression[82]. Therefore, our analysis adds to previous findings[72] by implicating a specific cell type—glial cells—in the generation of distinct behavioral states.

Taken together, these results offer important ground-truthing for the comparison of regulatory networks jointly inferred across samples and, more generally, demonstrate the applicability of SimiC to non-model organisms.

## Discussion

In this work we introduced SimiC, a gene regulatory network (GRN) inference method for single-cell RNA-Sequencing (scRNA-Seq) data. SimiC expects as input the linear ordering between different cell phenotypes (only if more than two phenotypes are considered). SimiC infers a GRN per phenotype while imposing a similarity constraint that forces a smooth transition between GRNs of consecutive states. This allows us to compare GRN architectures between distinct phenotypic states, with far-reaching implications for systems biology. In contrast, other GRN inference methods such as SINCERITIES[24], ICAnet[28] or SCENIC[26]

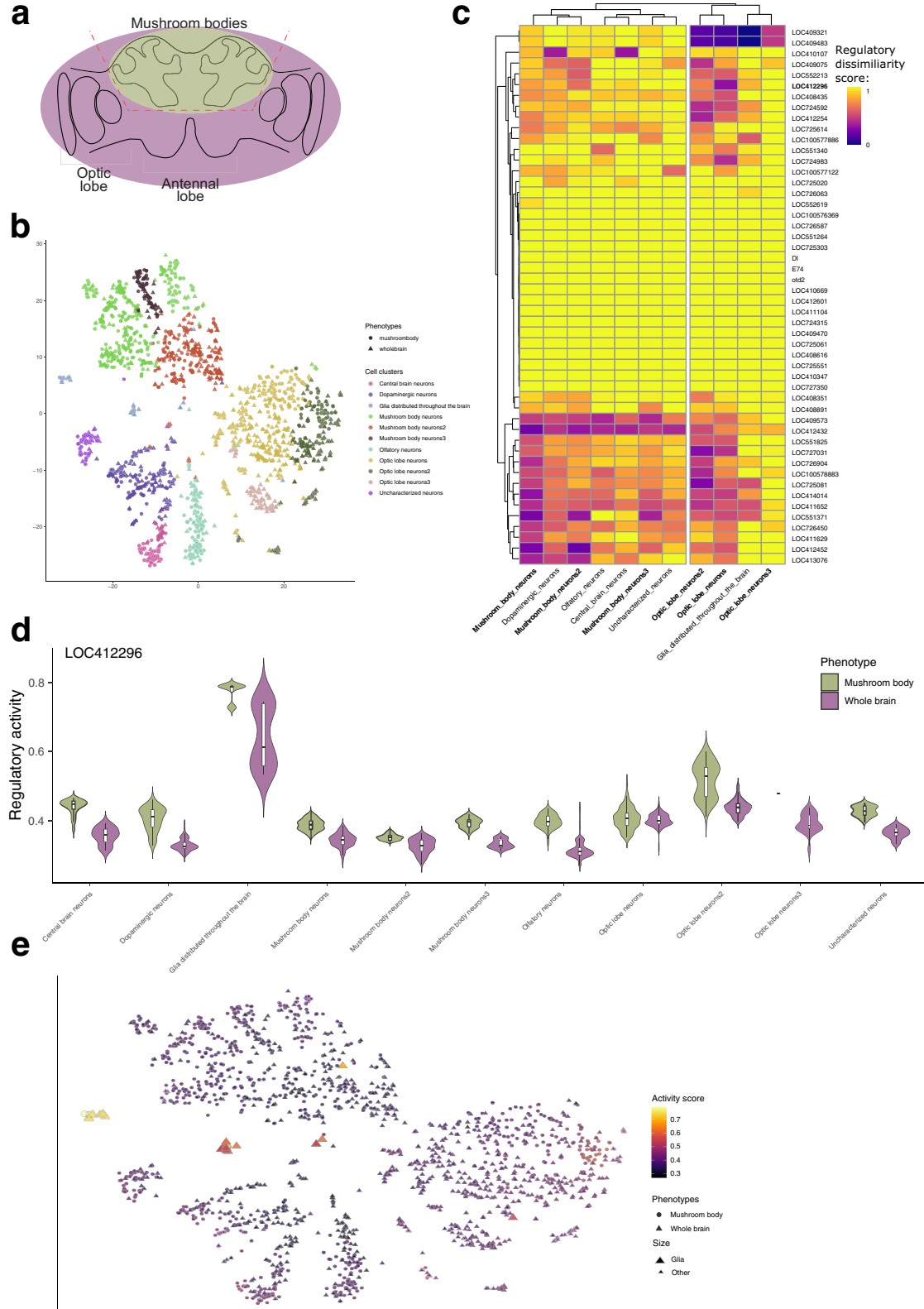

**Fig. 7 Additional results of SimiC on the bee brain dataset containing cells coming from either whole-brain (WB) or mushroom body (MB)**[70]**. a** Scheme depicting the base anatomy of the bee brain, and the line (dashed red) marking the surgical separation of the MB. **b** tSNE visualization of the cells colored by cell state and shaped by the phenotype. **c** Heatmap depicting the regulatory dissimilarity score across the whole brain and the mushroom body, for each cell state. **d** Violin plots showing the regulatory activity of the regulon LOC412296 on the different cell states, on the mushroom body, and whole brain. **e** tSNE visualization of the bee brain cells colored by the regulatory activity score of the regulon LOC412296. Shapes correspond to the two studied phenotypes and the size highligths the cells belonging to the glia.

do not impose any similarity constraints, and are therefore either applied to the whole scRNA-Seq dataset, generating a single GRN for all cell phenotypes, or to each phenotype independently, potentially missing relationships between cells from different phenotypes. We showed that jointly inferring phenotypically distinct GRNs can uncover regulatory relationships (or dissimilarities) that would have otherwise been missed, thus establishing a new approach to quantitating differences or dissimilarities between the GRNs of distinct cellular phenotypes. Another advantage of SimiC with respect to SCENIC and ICAnet is that the inferred regulatory relationships between target and driver genes are accompanied by a weight, indicating the strength and direction of the regulation, and hence providing an extra layer of resolution. For example, factoring in the weights allowed us to recapitulate known activators and repressors of key target genes on CAR T cell therapy. In addition, we showed that SimiC's computed scores uncover phenotype- or state-specific clusters, unlike those of SCENIC and ICAnet. Finally, SimiC works well across a range of systems, both model and non-model alike, and does not require detailed annotations to the transcriptome, whereas other methods rely on prior knowledge of specific features like TF-binding motifs.

In summary, we provided significant evidence to show that SimiC has the potential to reveal complex gene regulatory relationships across different phenotypes or timepoints, for model and non-model organisms. Furthermore, we anticipate that SimiC will be applicable to advancing our understanding of the relationship between brain transcriptomic state and behavioral variation, and GRNs have been shown to be a reliable surrogate of complex behavioral phenotypes[5,6]. SimiC builds on these previous studies by allowing for a direct, quantitative comparison of GRN structure between behavioral states. The key of SimiC is its optimization function that jointly infers the GRNs that model each phenotype.

## Methods

In what follows we describe SimiC in more detail, both the method and the evaluation metrics, and provide details on the considered datasets, including any pre-processing steps to generate the input data needed by SimiC.

**Notation.** scRNA-Seq data are expressed as a matrix of dimension number of sequenced cells by number of measured genes. We further classify genes as driver or target genes, and use notation $X$ and $Y$ to represent their expression matrices, respectively. If there are $c$ cells, $m$ driver genes, and $n$ target genes, $X \in \mathbb{R}^{c \times m}$ and $Y \in \mathbb{R}^{c \times n}$. We use upper case notation $X_i \in \mathbb{R}^c$ ($Y_i \in \mathbb{R}^c$) to represent the expression of driver (target) gene $i$ across cells, and lower case notation $x_i \in \mathbb{R}^n$ ($y_i \in \mathbb{R}^m$) to represent the expression of cell $i$ across target (driver) genes. That is, $X_i$ is the $i$th column of matrix $X$ (analogously for $Y$), and $x_i$ is the transpose of the $i$th row of matrix $X$ (analogously for $Y$).

**Feature selection.** In order to apply the algorithm more efficiently and filter out the least informative genes for better robustness, we first perform a feature selection on the gene space. Specifically, we select separately the driver and target genes with the highest median absolute deviation (MAD) from their mean expressions. The MAD value is a measure of the variability across samples that is robust to outliers. Specifically, for a given gene expression profile $X_i$ (or $Y_i$), its MAD value is given by $MAD(X_i) = \text{median} (|X_i - \bar{X}_i|)$, where $\bar{X}_i$ is the average value of $X_i$. The expression of a gene is more scattered if its MAD is larger and vice versa. Note that this type of filtering is typically performed in single-cell data analysis, where sometimes other metrics such as variance are used instead of the MAD value.

**GRN inference background.** In the general setting of bipartite GRN inference, given a set of $m$ driver genes and a desire set of $n$ target genes, the goal is to find a weighted bipartite graph between these two sets, where the weights describe the regulation activity of gene pairs. The common assumption is that the expression of the target genes of a cell $i$, denoted as $y_i \in \mathbb{R}^n$, can be approximated by a linear combination of its driver genes, denoted as $x_i \in \mathbb{R}^m$, under a Gaussian noise assumption, i.e., $y_i = W^T x_i + b + \epsilon_i$. Here the matrix $W \in \mathbb{R}^{m \times n}$ is the incidence matrix between driver and target genes, with the $j$th column $W_j$ being the connectivity strength between target gene $j$ and the set of driver genes. For ease of notation, we include the bias term $b \in \mathbb{R}^n$ into $W$ by extending $x_i$ to $[x_i, 1] \in \mathbb{R}^{m+1}$. It should be noted that assuming linear dependencies for the inference of GRNs is quite established in the field[3,7,29]. In

addition, methods relying on more complex non-linear modelings are computationally intensive and cannot generally handle more than 1000 genes[29].

Given a group of independent and identically distributed (i.i.d.) expression profiles, the common approach[7] is to minimize the approximation error $\frac{1}{2}|Y - XW|^2$. The solution to such least squares problem usually results in a dense incidence matrix $W$. Yet, in practice, it is believed that only a subset of driver genes regulate a given target, and hence the connection in the graph should be sparse[3,7]. In order to have a more robust model and sparse incidence matrices, the most common optimization problem for GRN inference is expressed as:

$$\min_{W \in \mathbb{R}^{m \times n}} f(W) = \min_{W \in \mathbb{R}^{m \times n}} \frac{1}{2}|Y - XW|^2 + \lambda|W|_1 \qquad (1)$$

This is a LASSO formulation, where $Y$ is the target expression matrix and $X$ is the driver expression matrix[83]. Such approach is effective when the expression matrix is composed of different samples of bulk sequencing profiles, which can be treated as i.i.d. samples. However, in the case of single-cell RNA sequencing, this assumption is no longer true. In addition, different cell states are expected to be governed by different GRNs, which correspond to different incidence matrices. To account for this, one approach would be to apply LASSO independently on each cell type. However, we would lose the information from the other cell states, which might be useful given that the cells generally come from the same region (or tissue) and a linear ordering may exist between them. In other words, in most scenarios the underlying regulatory networks of the different cell types are expected to share some common functions due to the asynchronous cell progression, which translates into some level of similarity between the corresponding incidence matrices.

**Data imputation.** Due to the limitations of current single-cell sequencing technologies, the raw scRNA-Seq datasets (i.e., the raw expression matrices) are often extremely sparse[12]. However, it is possible to impute the missing expression values by using combined information from all the sequenced cells. Thus, SimiC uses imputed data for the inference of GRNs. We use MAGIC[12] for this purpose. MAGIC was chosen as it has been shown to be the imputation method that most consistently outperforms the other ones[84]. Nevertheless, SimiC can be used with data imputed with any available imputation method (Supplementary Fig. 9). It should be noted also that imputation methods generate imputed data of varying levels of sparsity (Supplementary Table 3 and Fig. 9). The conducted experiments suggest that SimiC is more affected by the chosen imputation method than by the resulting sparsity level.

**SimiC optimization algorithm.** With the imputed scRNA-Seq expression data, the cell state labels for each cell, and the associated ordering, our optimization problem for GRN inference is defined as:

$$\min_{W^k, k \in [1:K]} f(W^1, W^2, \cdots, W^K) = \min_{W^k, k \in [1:K]} \sum_{k=1}^{K} \frac{1}{2}|Y^k - X^k W^k|^2 + \sum_{k=1}^{K} \lambda_1|W^k|_1 + \sum_{k=1}^{K-1} \lambda_2|W^k - W^{(k+1)}|_2^2, \quad (2)$$

where $W^k$ is the incidence matrix of the GRN that we want to infer for cell state $k$, $K$ is the number of states, $Y^k$ is the target expression matrix of cells in state $k$, and $X^k$ is the corresponding driver expression matrix. Assuming $n$ target genes and $m$ drivers, and $s_k$ cells under cell state $k$, the dimensions of $Y^k$, $X^k$, and $W^k$ are $s_k \times n$, $s_k \times m$, and $m \times n$, respectively. Note that the dimension of $X^k$ and $Y^k$ may change across different states, but the dimension of the incidence matrices is always the same. The reason is that the GRNs for different cell states share same set of nodes (driver and target genes), and only differ in the edge weights. The first summation in our objective with the $\ell_1$ regularization term, $\sum_{k=1}^{K} \lambda_1|W^k|_1$, serves the same purpose as in LASSO, i.e., it controls the sparsity of the incidence matrices. With or without the cell states ordering, minimizing the first part (i.e., setting $\lambda_2$ to zero) is equivalent to solving LASSO for every state independently.

The second regularization term, $\sum_{k=1}^{K-1} \lambda_2|W^k - W^{(k+1)}|_2^2$, is the *similarity constraint*. As mentioned above, we would like to smooth the GRNs transition process (we assume in the formulation that the cell states $[1:K]$ are linearly ordered). With the order of cell states given, it is reasonable to assume that two consecutive states should share common edges. This translates into minimizing the pairwise difference of the corresponding GRNs, so as to maintain the common graph structure among them. Note that adding a second order regularization term will tend to make the incidence matrices denser. The trade-off between adding the sparsity constraint and the similarity constraint is controlled through the values of $\lambda_1$ and $\lambda_2$. For example, in cases in which the cell states are well separated, the smoothness assumption of the GRNs is weaker and hence $\lambda_2$ should be smaller, and vice versa.

**Algorithmic implementation.** Note that our objective function is convex on $[W^1, ..., W^K]$, but not smooth due to the existence of the $\ell_1$ norm regularization term. To solve the optimization problem, we use a random block coordinate descent (RCD) algorithm, summarized in Algorithm 1, where $W(t)$ indicates the incidence matrix $W$ at iteration $t$ and $\gamma_k$ is chosen to be the largest eigenvalue of $X^{k^T} X^k$ for each cell state.

---

**Algorithm 1:** RCD for SimiC

---

**Input:** $X^k, Y^k, \gamma_k$, for $k \in [1, \ldots, K]$, $\lambda_1, \lambda_2$, number of iterations $T$;

**Initialization** $: W^k(0) = \mathbf{0}, \forall k \in [1, \ldots, K]$;

**for** $t \in [1, \ldots, T]$ **do**

 Randomly pick a cell state $l \in [1, \ldots, K]$

 $\widetilde{\nabla}_{W^l} f = \frac{2}{m} X^{l^T}(X^l W^l(t-1) - Y^l) + \lambda_1 sign(W^l(t-1))$

 $\nabla_{W^l} f = \widetilde{\nabla}_{W^l} f + 2\lambda_2 \left(2W^l(t-1) - W^{l+1}(t-1) + W^{l-1}(t-1)\right)$

 $W^l(t) = W^l(t-1) - \frac{1}{\gamma_l}\nabla_{W^l} f$

**end**

---

The hyper-parameter $\lambda_1$ controls the sparsity of the $W^k$ matrices, whereas $\lambda_2$ controls the inter-matrix dependencies and thus can be tuned based on the underlying structure of the cell states. We choose them from the polynomial set $\{10^{-5}, 10^{-4}, 10^{-3}, 10^{-2}, 10^{-1}\}$. Specifically, $\lambda_1$ and $\lambda_2$ are chosen using five-fold cross validation, where the dataset is randomly split into training and validation sets in a proportion of 80:20. We evaluate the approximation performance by the average adjusted $R^2$ value on the left out validation sets. This process is done in each pair of combination of $\lambda_1$ and $\lambda_2$ from the pre-selected polynomial set and the ones resulting in the highest average adjusted $R^2$ are chosen as the final values of $\lambda_1$ and $\lambda_2$. The output of Algorithm 1 is a group of incidence matrices $[W^1, \ldots, W^K]$, each corresponding to the GRN of one state. The $W$ matrices all have the same dimension $m \times n$ (number of driver genes by number of target genes) and the same column/row index for the corresponding driver-target pairs, i.e., entry $W_{i,j}$ is the weight between the $i$th driver and the $j$th target genes.

**Activity score**. To better understand how the activity changes in the cell population, we propose to use a new metric: the activity score. Similar to[26], we compute one activity score per driver gene and per cell. The input to the activity score workflow is the expression profile of the cell and the regulon (i.e., the target genes connected to the given driver gene along the weights for each edge), and the output is the relative activity of this regulon in the cell. Note that in the objective of Eq. 2, the weights are directly comparable across drivers for a given target gene, but not across targets for a given driver since the expression for different targets may vary. Hence, we first normalize the incidence matrices $W$ for each driver gene (i.e., each row) by dividing each element by the norm of the corresponding target expression. Next, for each cell, we order the normalized weights by the expression of the target genes in the cell. We denote the resulting normalized and ordered weight matrix by $\widehat{W}$. We then compute the cumulative sum of the ordered weights. The activity score of the considered driver gene in the cell is then defined by the normalized area under the cumulative sum curve, which can be expressed as:

$$\text{Activity Score (driver gene } i) = \frac{\sum_{t=1}^{T}\left(\sum_{n \leq t} \frac{1}{T}\widehat{W}_{i,n}\right)}{\sum_{t=1}^{T}\widehat{W}_{i,t}},$$

where $T$ is the number of target genes in the regulon $i$. Note that the activity score is a score between 0 and 1. When the larger weights are ranked higher (i.e., the corresponding target genes are highly expressed in the cell), the numerator is larger and the activity score will be closer to 1, which shows that the driver gene under consideration is more active in that particular cell. On the contrary, when the larger weights are ranked lower, the numerator gets smaller and the activity score gets closer to 0. The activity score is hence a comprehensive measure that takes into account both the gene expression and the weighted regulon structure information. More importantly, it can be computed for every cell and every driver gene, which gives a higher precision measure of a driver's activity within the cell population. The activity scores are stored in a matrix $A \in \mathbb{R}^{c \times m}$, where $c$ is the number of cells and $m$ is the number of driver genes. We compute the activity score matrix $A^k$ for each cell state $k \in [1:K]$ based only on the expression of the cells in that state and the corresponding incidence matrix $W^k$.

**Regulatory dissimilarity computation**. Given a driver gene $i$ of interest and the activity score matrix for state $k$, $A^k$, we can compute the distribution of activity scores for all cells in state $k$. We can repeat this process for all $K$ states, generating a set of $K$ distributions for a given driver gene. Then, by analyzing the changes in distribution across two states, we can infer whether the driver gene in question plays a role in the state transition. On one hand, when the two distributions are "separated" (e.g., they have a non-overlapping support), the activity of the weighted regulon has shifted between states. This suggests that the driver gene may be highly correlated with the state transition. On the other hand, when the two distributions are similar, the regulon activity remains unchanged in both states. This suggests

that the driver gene is less likely to have an influence in the state transition. To formally define the variation of the weighted regulon activity across multiple states, we use the total variation (TV) distance. The TV of two probability measures $P$ and $Q$ on a countable sample space $\Omega$ is defined as:

$$\delta(P, Q) = \frac{1}{2}\sum_{\omega \in \Omega}|P(\omega) - Q(\omega)| \tag{3}$$

Let $\{P_1, \ldots, P_K\}$ denote the set of activity score distributions. We then consider the following minmax version of total variation for multiple distributions:

$$\delta_{\text{minmax}}(\{P_1, \ldots, P_K\}) = \frac{1}{K}\sum_{\omega \in \Omega}\left(\max_{P_i} P_i(\omega) - \min_{P_j} P_j(\omega)\right) \tag{4}$$

It can be easily verified that this metric has values between 0 and 1, with values closer to 0 when the group of distributions is more similar to each other, and closer to 1 when the group of distributions is more divergent. The proposed minmax TV focuses on the outline of all distributions jointly, and it takes a higher value when the distributions are more disjoined than when they overlap with each other. Note that when $K = 2$, this metric degenerates back to the original definition of the total variation.

**Filtering of regulons**. In order to remove driver genes with very little weight assigned to target genes (due to numerical imprecisions on the optimization algorithm), we apply a filtering based on the bayesian information criterion (BIC). Specifically, SimiC only keeps the smaller set of driver genes needed to model at least 90% of its expression variance.

Thus, for each target gene, we order the driver genes by their associated weight and compute the squared sum of all the driver genes' weights. We then select the driver genes with larger weight until the squared sum of their weight is greater than the 90% of all the weights squared. Mathematically, for target gene $j$, we define the score $S_{ij}$ as:

$$S_{ij} = \frac{\sum_{k=1}^{i} W_{kj}^2}{\sum_{l=1}^{L} W_{lj}^2},$$

where $L$ is the number of driver genes connected to target gene $j$. We then find the smallest $i$ that satisfies $Sij \leq 0.9$, and select the first $i$th driver genes (ordered by decreasing weight).

In addition, for each target gene, we measure the adjusted $R^2$ between its true gene expression and the inferred expression using the driver genes' expression and the inferred GRN. If the adjusted $R^2$ is below 0.7, we consider that the target gene did not obtain a good fit, and it is therefore discarded. This step is performed to account for false positives in the inference process.

**Generation of synthetic GRNs and the corresponding scRNA-Seq data**. We generated a synthetic dataset with 5 states. The GRN for the first state is generated randomly, with the weight of an edge being equal to $+1$ with probability 0.1, to $-1$ with probability 0.1, and to 0 with probability 0.8. To generate the GRN of the next state, we start with the GRN of the previous state and randomly modify each of the edges with a conditional probability that depends on the previous state's network. Specifically, for a nonzero value edge, we change its sign with probability 0.3 and we set the weight to 0 with probability 0.7. For zero-value edges, we keep the same value with probability 0.5 and change it to $+/-1$ with an equal probability of 0.25. Note that this scheme keeps consecutive incidence matrices at similar sparsity levels. We do this procedure for each state, sequentially. Finally, the target gene's expression is generated by sampling from a negative binomial distribution with $n = 5$ and $p = 0.5$.

**Computation environment**. For all performed experiments, the SimiC framework, including the computation of the regulon activity scores, was run on a Linux server with a 80 cores Intel Xeon CPU and 512GB of memory.

On the synthetic dataset, the whole SimiC framework took 7 min to run using 16 cores and 5GB of RAM. More specifically, the optimization problem was solved in 4 min, and the regulon activity scores were computed in 3 min. Note that the running time of SimiC grows roughly linearly with the number of cells, number of driver genes, and number of target genes. For comparison, SINCERITIES took around 2 min to infer the GRNs for the same dataset.

For the CAR T cell dataset, which contains a total of 16K cells, the inference of the GRNs (with 100 driver genes and 1000 target genes) took 32 min. On the other hand, SCENIC took around 9 min and ICAnet 132 min, both run on the whole data.

**Authority and hub scores**. To better understand how the genes are modulated, we represent them as a weighted network. This network is created with the R library *iGraph*, where the nodes are genes and the edges are the weights computed by SimiC. These weights represent the strength of the modulation, while the sign represents the polarity. To represent the changes over the different phenotypes, we compute the Kleinberg's authority and hub centrality scores[53] for each node on each phenotype, and the statistical comparison was assessed by a Wilcoxon signed-rank test.

**ChIP-Seq analysis**. In order to asses the goodness of the driver-target genes connectivity, we validate these connections with empirical binding data. To perform this analysis, we used the database provided by "Integrative ChIP-Seq analysis of regulatory regions (ReMap2020)"[39], which contains transcriptional regulators peaks derived from curated experiments coming from Human samples. The peaks were annotated using the R library *ChIPpeakAnno*.

The odds ratio and hypergeometric test were calculated for each driver gene, by locating on the database all driver genes' binding sites and annotating them to target genes being located at most 5Kbps upstream the binding site.

**Reporting summary**. Further information on research design is available in the Nature Research Reporting Summary linked to this article.

## Data availability

The simulated data generated in this study is available at https://doi.org/10.13012/B2IDB-4996748_V1. The analyzed scRNA-Seq data composed of monocyte cells and CD4+ T-lymphocytes have GEO accession ID GSE139369. The analyzed scRNA-Seq data of CAR T cells have GEO accession ID GSE125881. The analyzed scRNA-Seq data of hepatocytes have GEO accession ID GSE151309. The analyzed scRNA-Seq data of honey bees have GEO accession ID GSE130785. A description of the datasets is provided in Supplementary Table 3.

## Code availability

SimiC is open source (MIT license) and available at https://github.com/jianhao2016/SimiC, together with installation instructions and scripts to run it either from Python or R. In addition, we have made available the data needed to run the tutorial at https://doi.org/10.13012/B2IDB-3975180_V1. The code used to run the different methods is described in Supplementary Notes 1–4.

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

## Acknowledgements

H2020 Marie S. Curie IF Action, European Commission, Grant Agreement No. 898356 (M.H.). National Institutes of Health (NIH) R01HL126845, R01AA010154 (U.V.C., S.B., A.K.). Planning Grant Award from the Cancer Center @ Illinois (U.V.C., S.B., A.K.). NIH Tissue microenvironment training program, T32-EB019944 (SB). Scott Dissertation Completion Fellowship, UIUC (SB). Instituto de Salud Carlos III (ISCIII) PI20/01308 (T.E., F.P.). CIBERONC CB16/12/00489, co-financed with FEDER funds (F.P.). Investigador AECC award, "Asociación Española Contra El Cáncer", INVES19059EZPO (TE). The Accelerator award CRUK/AIRC/AECC joint funder-partnership, EDITOR (FP). Gipuzkoa Fellows from the Basque Government, Spain (I.O.). Ramon y Cajal grant, Spain (I.O.). Gobierno de Navarra, Departamento de Salud and Departamento de Industria, Proyecto Estrategicos, DESCARTHES and AGATA (MEC, JRR). The authors would like to thank Dr. Josepmaria Argemi for insightful discussions.

## Author contributions

Conceptualization: J.P., G.S., I.O., and M.H.; methodology: J.P., G.S., I.O., and M.H.; software: J.P. and G.S.; formal analysis: G.S. and M.H.; investigation: J.P., G.S.; validation: I.M.T., M.E.C., U.V.C., S.B., T.E., and J.R.R.; supervision: A.K., F.P., I.O., M.H., and J.R.R.; writing-original draft: J.P., G.S., I.O., and M.H.; visualization: G.S., I.M.T., M.E.C., and M.H.; writing-review & editing: all authors.

## Competing interests

The authors declare no competing interests.
