## [Peer Review File · Communications Biology]

Reviewers' comments:

Reviewer #1 (Remarks to the Author):

In this manuscript, the authors introduced a new single-cell gene regulatory network (GRN) inference method, named SimiC. The input include: (1) imputed single cell expression data, (2) a list of driver genes, and (3) cell labels and (time) ordering information. The method is based on the fused LASSO regression technique to infer the GRNs. The method was compared against SCENIC, SINCERITIES, Seurat, and ICAnet. The performance of each GRN inference method was assessed using both simulation and 2 real datasets. The link to the simulated data is not active anymore, and I was not able to run the analysis using the script provided. Also, the results provided in the manuscript show little improvement over other methods. Below are my comments, which I hope it would help to improve their article.

1/ I was able install the package but could not run the script provided. I encountered errors using different machines (Windows and Linux). Therefore, I was not able reproduce the analyzed results presented in the manuscript. Also, script for running other methods is cutoff. I suggest the authors to check the package and scripts carefully. The authors should provide a Docker image for reproducibility purposes.

2/ In the method section, the authors assume that driver genes and target genes are linearly dependent. Is there any evidence that can be used to support this assumption? The threshold for selecting driver genes and target genes is not well justified. Do the results change if the authors change these thresholds.

3/ The authors did not describe how they simulated the synthetic data. What software did they use to construct the ground truth network? The link to download the simulation data is no longer accessible.

4/ The evaluation is weak with only 1 simulated dataset and 2 real datasets. Even in this case, the performance of SimiC is comparable to that of existing methods. The authors should add more real and simulated datasets to their analysis.

5/ The number of methods compared is low. The authors compared their method against 2 methods for synthetic data, and 4 methods for real data. There are many other methods available, including the 16 methods listed in a recent survey (PMID: 33057606).

6/ In the method pipeline, SimiC uses MAGIC to impute single-cell expression. Is SimiC still superior when using other imputation methods using the exact same data analyzed in the manuscript?

7/ Dropout rates are known to impact the performance of any analysis method, including GRN inference methods. I suggest the authors to test their software with different dropout rates, for both simulations and real datasets.

Reviewer #2 (Remarks to the Author):

In this paper, the authors developed a single-cell gene regulatory inference framework called SimiC by jointly learning distinct gene regulatory dynamics per phenotype. This is a method paper and the tool is helpful to the research community in single cells. The following lists some comments.

First, in SimiC, the authors firstly applied Lasso regression to obtain the co-expression matrix like GENIE3 in SCENIC. However, there may be many false positives in regulons. A control step is expected to enhance the inference ability.

Second, in algorithm1, what's the meaning of parameter m in the 'for' loop. More details need be introduced in the main text for the algorithm.

Third, for each phenotype, SimiC reconstructs a GRN. In real scenario, some phenotypes may have little samples compared with some others. Whether the number of the samples have a big effect to the results. This need be checked.

Fourth, the comparison study. It is available for some other methods/strategies. It is good to do the comparison studies in Figure 5 and Figure 6. While the results need be introduced in details, especially the ARI metrics. It is expected to introduce more specific metrics to highlight the comparison study. The authors can focus on the feature of SimiC e.g., across multiple cells.

Point-by-point Response to Reviewers

Communications Biology manuscript COMMSBIO-21-1733-T

First, we would like to thank you for sending our manuscript for peer review, and the reviewers for the time taken to review our manuscript. Below we provide a point-to-point response to the reviewers. Please note however that this submission is not intended as a fully revised version of the manuscript based on the reviewers' comments, but as a rebuttal to get the opportunity to carefully address all the raised concerns in a fully revised version of the manuscript. Nevertheless, we have already addressed most reviewer comments and modified the paper and Supplementary data accordingly.

Reviewer #1

In this manuscript, the authors introduced a new single-cell gene regulatory network (GRN) inference method, named SimiC. The input include: (1) imputed single cell expression data, (2) a list of driver genes, and (3) cell labels and (time) ordering information. The method is based on the fused LASSO regression technique to infer the GRNs. The method was compared against SCENIC, SINCERITIES, Seurat, and ICAnet. The performance of each GRN inference method was assessed using both simulation and 2 real datasets. The link to the simulated data is not active anymore, and I was not able to run the analysis using the script provided. Also, the results provided in the manuscript show little improvement over other methods. Below are my comments, which I hope it would help to improve their article.

Comment 1.1 I was able install the package but could not run the script provided. I encountered errors using different machines (Windows and Linux). Therefore, I was not able reproduce the analyzed results presented in the manuscript. Also, script for running other methods is cutoff. I suggest the authors to check the package and scripts carefully. The authors should provide a Docker image for reproducibility purposes.

Response: We are sorry to hear the arisen issues while running SimiC. We did run the code in different machines to detect any possible issues with its installation, but, as evidenced by the problems faced by the reviewer, we failed to address all cases. Thus, we have modified the GitHub repository to make the installation self-contained (not dependable on external libraries) and the software executable in one command (to facilitate its use), as well as generate a docker container (as suggested by the reviewer). In addition, we have included a step-by-step tutorial that reproduces the results of the paper.

We would like to note that in this regard there is a very recent paper (Nature Methods: <https://www.nature.com/articles/s41592-021-01256-7>) in which they suggest levels of reproducibility standards for Machine Learning in Life Sciences (Table 1 of that paper). The new GitHub code follows the Gold Standard (the first submitted code lay between bronze and silver standard).

Finally, regarding running the other methods, all commands were provided in the Supplementary.

Comment 1.2 In the method section, the authors assume that driver genes and target genes are linearly dependent. Is there any evidence that can be used to support this assumption? The threshold for selecting driver genes and target genes is not well justified. Do the results change if the authors change these thresholds.

Response: Linear dependencies in GRNs have been a wide assumption when developing methods to infer GRNs on large datasets. For example, in the review paper pointed out by the reviewer (PMID:

33057606), the best performing GRN inference methods are based on correlation (i.e., linear relationship, PMID: 33057606 Figure 4). Moreover, current state-of-the-art GRN inference method for bulk RNA-Seq data (PMID: 29331675 (2018), 31287491 (2020)) also assume linear relationships. Note that methods relying more complex non-linear modelings (such as those relying on ordinary differential equations) are computationally intensive and generally cannot handle more than 1,000 genes (see review paper PMID: 33057606, Figure 6). We have elaborated on this in the paper.

Regarding the threshold mentioned by the reviewer, there is no threshold to select drivers and targets. As stated in the paper, driver genes were selected according to the Human Protein Atlas annotation (datasets 1-3), and the Honeybee Atlas annotation (dataset 4). If the reviewer refers to how we select the subset of target and TFs to run the different methods, we used the ones with the highest Mean Absolute Deviation (MAD) value. Note that this type of selections is generally done in single-cell data analysis, where sometimes other metrics, such as the variance, are used instead of the MAD value. We made sure to make this clearer in the text.

Comment 1.3 The authors did not describe how they simulated the synthetic data. What software did they use to construct the ground truth network? The link to download the simulation data is no longer accessible.

Response: We are sorry that this was not clearer in the text. Figure 2 shows how the simulated data is generated. Moreover, we describe it in more detail in the Methods section (see “Generation of synthetic GRNs and the corresponding scRNA-Seq data”). We have made it clearer in the text.

As for the access to the synthetic data, we have deposited all in the Illinois Data Bank (<https://databank.illinois.edu/>) to ensure its availability. The link for the data access is https://doi.org/10.13012/B2IDB-4996748_V1. The link has been corrected in the text (the previous one was temporal).

Comment 1.4 The evaluation is weak with only 1 simulated dataset and 2 real datasets. Even in this case, the performance of SimiC is comparable to that of existing methods. The authors should add more real and simulated datasets to their analysis.

Response: We kindly disagree with the reviewer. We are currently using 4 real datasets and 1 simulated one. Importantly, GRN inference validation is a hard problem (as we stated in the introduction, and as stated in the review paper pointed by the reviewer [PMID: 33057606, Fig.1]). Hence, GRN inference methods generally use simulated data or simplified quantitative measures (such as the ARI score) to compare against existing methods (see for example ICAnet, Nucleic Acid Research, 2021, PMID: 33619563). In our paper, we took a step further and went deeply in the inferred biology by different methods to benchmark the performance of the methods, with the complexities that this brings. Also important is to note that other methods like SCENIC cannot be run in non-model organisms, contrary to our proposed method SimiC, as shown with the honeybee dataset that we used.

Regarding the performance differences, SimiC achieves a 16% and 60% improvement in clustering performance with respect to SCENIC (Figures 5H and 6H), the most widely used GRN inference method and the one achieving the best performance in the review paper pointed out by the reviewer (Brief, in Bioinformatics, 2020, PMID: 33057606). In addition, we improved by 16% the performance of ICAnet (Figure 6H), the most recent GRN inference method (Nucleic Acid Research, 2021, PMID: 33619563), while performing equally well on another dataset (Figure 5H). In comparison, the authors of ICAnet showed a 0-10% range of improvement with respect to SCENIC.

Nevertheless, we have included more analysis and quantitative assessments. Specifically, we have test the methods in 4 different hematopoietic differentiation trajectories, and assess the capabilities of the different methods to distinguish between differentiating cells. We show that SimiC is the only method that can reliably differentiate cells using a consistent set of regulons. On the other hand, SCENIC and ICAnet, when run independently on each cell-type, do not produce a consistent set of TFs and regulons that can be assessed

across the trajectory. When inferred jointly on the whole set of cells, SCENIC and ICAnet do produce a consistent set of regulons across all cells, but these regulons fail to distinguish across differentiating cells (see Figure at the end of this document).

We have added these results to the paper, and the figure to the Supplementary.

Comment 1.5 The number of methods compared is low. The authors compared their method against 2 methods for synthetic data, and 4 methods for real data. There are many other methods available, including the 16 methods listed in a recent survey (PMID: 33057606).

Response: We kindly disagree with the reviewer. While the reviewer points to a paper stating that there are 16 methods already proposed, after careful inspection, only 6 of them can handle more than 2,000 genes, and only SCENIC generates a set of regulons that allows the user to assess dynamics at the single-cell level. From the other methods handling more than 2,000 genes: SCIMITAR is no longer maintained and we were not able to run it; SCODE is designed to work with TF-TF interactions and is not recommended for full GRN inference¹; LEAP generates a gene-gene correlation matrix at specific “lags”, and thus, makes it unfit to generate regulon activity scores across cell-types. Further, SCENIC was the best performing one in all metrics.

In addition, we also included ICAnet (Nucleic Acid Research, 2021), a very recently proposed method that it is not included in the review paper pointed out by the reviewer. We would like to note also that this is not a review paper, and hence we chose to include the best performing methods (as generally done).

We have extended the introduction to comment on this more explicitly.

Comment 1.6 In the method pipeline, SimiC uses MAGIC to impute single-cell expression. Is SimiC still superior when using other imputation methods using the exact same data analyzed in the manuscript?

Response: We tested several imputation methods, but we did not observe any difference in performance. Given that, we chose MAGIC, as it has been found to be one of the imputation methods that most consistently outperforms the other ones (see Genome Biology (2020), PMID: 32854757). Nevertheless, if the reviewer considers it essential, we could include a comparison with other imputation methods, such as scImpute or SAVERx.

Comment 1.7 Dropout rates are known to impact the performance of any analysis method, including GRN inference methods. I suggest the authors to test their software with different dropout rates, for both simulations and real datasets.

Response: We would be happy to perform further drop-out testing. Note however that the 4 used real datasets contain different levels of drop-out. We have included this information in the paper: Supplementary Table S3.

Reviewer #2

In this paper, the authors developed a single-cell gene regulatory inference framework called SimiC by jointly learning distinct gene regulatory dynamics per phenotype. This is a method paper and the tool is helpful to the research community in single cells. The following lists some comments.

Comment 2.1 First, in SimiC, the authors firstly applied Lasso regression to obtain the co-expression matrix like GENIE3 in SCENIC. However, there may be many false positives in regulons. A control step is expected to enhance the inference ability.

Response: We thank the reviewer for this insightful comment. Indeed, a false positive control is included when inferring the regulons. Specifically, for each target gene we measure the adjusted R^2 between the true

¹Communication with the authors.

gene expression and the inferred expression using the TFs and the inferred GRN. If the adjusted R^2 is below 0.7, we consider that the target gene did not obtain a good fit, and it is therefore discarded. Note that an adjusted $R^2 > 0.7$ is a quite stringent threshold so that it controls the False Positive Rate (FPR). We are sorry that this was not clearer in the text. Hence, we have modified the manuscript accordingly to make this more explicit (see Methods).

Comment 2.2 Second, in algorithm1, what's the meaning of parameter m in the 'for' loop. More details need be introduced in the main text for the algorithm.

Response: m refers to the number of driver genes, as introduced earlier in the text. Nevertheless, to make it clearer, we have clarified the meaning of this parameter after Algorithm 1 in the new text.

Comment 2.3 Third, for each phenotype, SimiC reconstructs a GRN. In real scenario, some phenotypes may have little samples compared with some others. Whether the number of the samples have a big effect to the results. This need be checked.

Response: We did perform some simulations and saw that with as few as 100 cells SimiC could reconstruct reliable GRNs, which we believe is a reasonable number. Nevertheless, if the reviewer thinks this should be further checked, we can perform an additional analysis on this regard and include it in the revised manuscript.

Comment 2.4 Fourth, the comparison study. It is available for some other methods/strategies. It is good to do the comparison studies in Figure 5 and Figure 6. While the results need be introduced in details, especially the ARI metrics. It is expected to introduce more specific metrics to highlight the comparison study. The authors can focus on the feature of SimiC e.g., across multiple cells.

Response: Note that the comparison study presented in Figure 5 includes SimiC, SCENIC and ICAnet. Moreover, for the comparison of the ARI score, we also included Seurat, which is widely used in practice. Note that SINCERITIES was not included in this analysis as it needs at least four states (phenotypes) to run, as specified in the caption of Figure 5.

Regarding Figure 6, note that we did include for the comparison all considered methods, that is, SimiC, SCENIC, SINCERITIES and ICAnet. However, for ICAnet, none of the generated modules contained any of the analyzed TFs (YBX1, CEBPA or CEBPB). Hence is omission in panels E-G of Figure 6. We have now included in the caption that the modules generated by ICAnet did not contain any of the analyzed TFs, to clarify this point. Note however that ICAnet is included in the ARI score comparison, as well as Seurat (similarly to Figure 5).

Regarding the ARI score, we have now included a citation, as well as its full name (Adjusted Rand Index) rather than just the acronym. Finally, regarding the inclusion of more specific metrics, since there are a myriad of clustering metrics (Silhouette Coefficient, MI-based metrics, Dunn Index, etc.), if the reviewer could be more specific on which of these they find it more relevant, we would be happy to add them to the revised version.

Figure I: On the CD4+ T-lymphocytes cells [Granja et. al., 2019], we selected five known differentiation trajectories, namely: i) HSC -> Early Erythrocyte -> Late Erythrocyte, ii) HSC -> CLP -> NK, iii) HSC ->CLP -> B, iv) HSC -> CLP -> CD8+ CD4+, and v) HSC -> GMP -> Monocytes CD14+. For ease of notation, we refer to the states of a given trajectory as Early -> Medium -> Late. The tSNE plot for the original data shows the cells colored by their state (yellow, green, and purple, respectively), for each trajectory. We applied SimiC to the cells of each trajectory (using the corresponding ordering), and then clustered the cells with k-means ($k = 3$) using the generated regulon activity scores. The tSNE plots and the corresponding Rand Index (RI) scores (close to 1 in all cases) demonstrate that SimiC's computed scores can reliably differentiate cells. We did a similar analysis with SCENIC and ICANet, when run independently on each state and on the whole data (denoted as *w/o cell labels*). In the former case, we also show the overlap in the generated regulons (for SCENIC) or modules (for ICANet) across the three states. For clustering, we used k-means for SCENIC and Seurat for ICANet. As can be observed, when run independently on each state, SCENIC obtains high RI scores for all trajectories except the last one, where the score drops below 0.75. When no states are given to SCENIC, the scores drop significantly (most below 0.5). ICANet fails to provide scores that can cluster cells by their state, in both considered cases.

Reviewers' comments:

Reviewer #1 (Remarks to the Author):

I appreciate the authors' effort to address my comments. The quality of the manuscript has improved significantly. However, I still have major concerns regarding software package and analyses.

1/ I am still not able to install SimiC package from the author's git repository. Below are the error messages from Linux and Window machines:

Linux: see attached "Linux Error.txt"

Window: see "Window Error.txt"

I tried to build a Docker image using the script provided in the software package but it was not successful: see "Docker Error.txt"

2/ The authors have not completely addressed my comments 1.6 and 1.7 (as numbered in the rebuttal letter). Regarding comment 1.6, the authors still do not show the comparison of SimiC with other imputation methods. In comment 1.7, the authors have not assessed SimiC's robustness against a different ratio of dropout on both simulated and real datasets. The authors should add those analyses in the next revision.

Reviewer #2 (Remarks to the Author):

In the new version, most of my former comments have been addressed. I appreciate them. For me, this is a method paper and it is good open the source code (package will be better). The comparison study is very important. I suggest make the materials in the comparison study also open. I have no more critical comment, only with minor change of figures with higher resolution. Thanks.

Point-by-point Response to Reviewers

Communications Biology manuscript COMMSBIO-21-1733-T

Editor comments:

We, therefore, invite you to revise and resubmit your manuscript, taking into account the points raised. In particular, please ensure the source code for comparisons is made available for reproducibility purposes. Also properly test your software.

Please highlight all changes in the manuscript text file.

Response: The revised version of the manuscript includes additional results with different imputation methods, which also showcase the ability of SimiC to work with different dropout rates. We have also updated the GitHub repository, which now includes a Docker image that has been extensively tested to ensure correct installation and usage, as well as the scripts used to compare against other methods. We believe the current GitHub repository contains all necessary information for this work to be reproducible.

We believe the submitted revised version addresses all the reviewers' comments.

We have uploaded two copies of the main manuscript, with and without highlighting the changes.

Reviewers' comments:

Reviewer #1

I appreciate the authors' effort to address my comments. The quality of the manuscript has improved significantly. However, I still have major concerns regarding software package and analyses.

Comment 1.1 I am still not able to install SimiC package from the author's git repository. Below are the error messages from Linux and Window machines: Linux: see attached "Linux Error.txt" Window: see "Window Error.txt" I tried to build a Docker image using the script provided in the software package but it was not successful: see "Docker Error.txt"

Response: We have updated the Docker image and the GitHub repository. In particular, we have created a dockerfile available at GitHub and the Docker image in dockerhub at (<https://hub.docker.com/r/guisesanz/simic/tags>), allowing to run SimiC and the posterior analysis via docker or singularity on any machine.

Additionally, to ensure there are no additional errors, we have tested installing and running SimiC on different machines. Specifically, SimiC has been successfully installed on different machines with different OS versions (Ubuntu 18.04.5 LTS, Ubuntu 16.04.6 LTS, and CentOS 7.5.1804).

Comment 1.2 The authors have not completely addressed my comments 1.6 and 1.7 (as numbered in the rebuttal letter). Regarding comment 1.6, the authors still do not show the comparison of SimiC with other imputation methods. In comment 1.7, the authors have not assessed SimiC's robustness against a different ratio of dropout on both simulated and real datasets. The authors should add those analyses in the next revision.

Response: To show the impact that the imputation method can have on SimiC, we now include results with imputation methods SAVERX, scVI and scImpute, and compare the performance to MAGIC. For the analysis, all methods are run with their standard parameters, which can yield different sparsity values. The resulting imputed data is then used as input for SimiC. The analysis was performed with the data from the CD4+ T-lymphocytes cells (same data used in Figure S7). For each trajectory and imputation method, we computed the Rand Index (RI) scores which indicates how well SimiC’s computed scores can reliably differentiate cells. The results are shown in Figure S9 (see also below) and show that using MAGIC for imputation produces the best results (RI close to 1 for all cases). The imputation method that produces the lowest RI values is scVI. SAVERX and scImpute produce mixed results, depending on the trajectory. Our results align with previous review articles where MAGIC has been found to be one of the imputation methods that most consistently outperforms the other ones (see Genome Biology (2020), PMID: 32854757).

It should be noted also that the sparsity (i.e., percentage of entries in the matrix that are zero) obtained after imputation with each method varies. Specifically, the sparsity goes from 91,11% on the original data to 0% with SAVERX and scVI, to 18.35% with MAGIC, and to 72.89% with scImpute. The sparsity obtained with each method is also specified in Figure S9.

Regarding the dropout rates, we now indicate the dropout rate of each dataset (before and after imputation) on Table S3 (see also below). They go from 0% to 18.69%. Additionally, the results from Supplementary Figure S9 show the performance of SimiC with different dropout rates (from 0% to 72.89%) obtained with different imputation methods for the same dataset. For all dropout rates we show at least one case in which SimiC achieves a Rand Index (RI) score close to 1, which demonstrates that the proposed SimiC pipeline is able to work with different dropout rates. We acknowledge that these results are affected by both the original dropout rate and imputation method. Nevertheless, all together the presented results seem to suggest that the choice of imputation method has a larger effect on SimiC’s performance than the dropout rate.

We abstained from randomly adding zeroes to MAGIC’s imputed data as this would break MAGIC’s logic behind which values are not imputed, and thus, would not be a realistic representative of a properly-imputed dataset by MAGIC.

In the revised manuscript we have extended the ‘Data imputation’ subsection within the Methods section to reflect why MAGIC was chosen for the experiments, and to refer the reader to the added Figure S9, which shows that SimiC can be run with any imputation method (although MAGIC seems to work the best, as it is consistent with the literature). We also comment on the fact that the imputation method seems to have more effect on SimiC’s than the original or resulting sparsity level.

Reviewer #2

Comment 2.1 In the new version, most of my former comments have been addressed. I appreciate them. For me, this is a method paper and it is good open the source code (package will be better). The comparison study is very important. I suggest make the materials in the comparison study also open. I have no more critical comment, only with minor change of figures with higher resolution. Thanks.

Response: The code has been made available in GitHub to facilitate its use. In addition, we have created a Docker image to ease installation and usage (see reply to comment 1.1 above).

Regarding the comparison study, we agree that it would be useful for the users to have the scripts available. We have made the scripts available at GitHub, under the Tutorial folder (https://github.com/jianhao2016/SimiC/tree/master/Tutorial/Scripts_comparison).

REVIEWERS' COMMENTS:

Reviewer #1 (Remarks to the Author):

The authors have addressed most of my comments.